# A compositional neural code in high-level visual cortex can explain jumbled word reading

**Aakash Agrawal[1], KVS Hari[2], SP Arun[3]\***

[1]Centre for BioSystems Science & Engineering, Indian Institute of Science, Bangalore, India; [2]Department of Electrical Communication Engineering, Indian Institute of Science, Bangalore, India; [3]Centre for Neuroscience, Indian Institute of Science, Bangalore, India

**Abstract** We read jubmled wrods effortlessly, but the neural correlates of this remarkable ability remain poorly understood. We hypothesized that viewing a jumbled word activates a visual representation that is compared to known words. To test this hypothesis, we devised a purely visual model in which neurons tuned to letter shape respond to longer strings in a compositional manner by linearly summing letter responses. We found that dissimilarities between letter strings in this model can explain human performance on visual search, and responses to jumbled words in word reading tasks. Brain imaging revealed that viewing a string activates this letter-based code in the lateral occipital (LO) region and that subsequent comparisons to stored words are consistent with activations of the visual word form area (VWFA). Thus, a compositional neural code potentially contributes to efficient reading.

## Introduction

Reading is a recent cultural invention, yet we are remarkably efficient at reading words and even jmulbed wrods (*Figure 1A*). What makes a jumbled word easy or hard to read? This question has captured the popular imagination through demonstrations such as the Cambridge University effect (*Rawlinson, 1976*; *Grainger and Whitney, 2004*), depicted in *Figure 1A*. Reading a word or a jumbled word can be influenced by a variety of factors such as visual, phonological and linguistic processing (*Norris, 2013*; *Grainger, 2018*). At the visual level, word reading is easy when similar shapes are substituted (*Perea et al., 2008*; *Perea and Panadero, 2014*), when the first and last letters are preserved (*Rayner et al., 2006*), when there are fewer transpositions (*Gomez et al., 2008*), when word shape is preserved (*Norris, 2013*; *Grainger, 2018*). At the linguistic level, it is easier to read frequent words, words with frequent bigrams or trigrams as well as shuffled words that preserve intermediate units such as consonant clusters or morphemes (*Norris, 2013*; *Grainger, 2018*). Despite these insights, it is not clear how these factors combine, what their distinct contributions are, and more generally, how word representations relate to letter representations.

Here, we hypothesized that, viewing a string of letters activates a visual representation that is compared with the representation of stored words. To probe visual processing, we devised a visual search task in which subjects had to find an oddball target string among distractor strings. This task does not require any explicit reading and is driven by shape representations in visual cortex (*Sripati and Olson, 2010a*; *Zhivago and Arun, 2014*). An example visual search array containing two oddball targets is shown in *Figure 1B*. It can be seen that finding OFRGET is easy among FORGET, whereas finding FOGRET is hard (*Figure 1B*), showing that FOGRET is more visually similar to FORGET. This makes FOGRET easy to recognize as FORGET, whereas OFRGET is harder. Thus, the visual similarity of the jumbled words FOGRET and OFRGET to the original word FORGET

**\*For correspondence:**
sparun@iisc.ac.in

**Competing interests:** The authors declare that no competing interests exist.

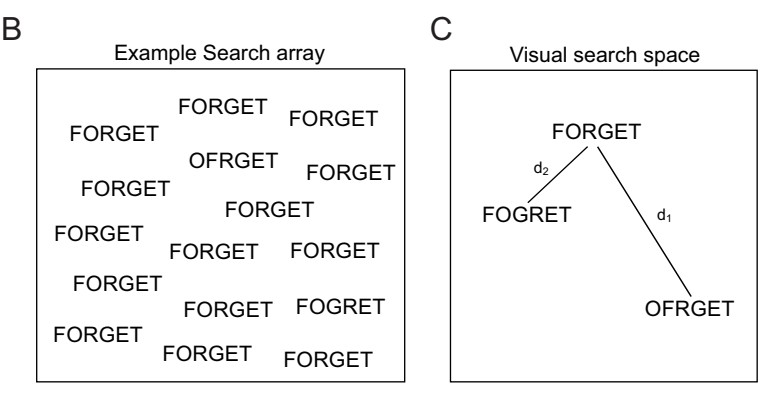

**Figure 1.** Reading jumbled words. (**A**) We are extremely good at reading jumbled words, as illustrated by the popular Cambridge University effect. (**B**) Visual search array showing two oddball targets (OFRGET and FOGRET) among many instances of FORGET. OFRGET is easy to find but not FOGRET. (**C**) Schematic representation of these strings in visual search space, arranged such that similar items (corresponding to harder searches) are nearby. Thus, FOGRET is visually more similar to FORGET compared to OFRGET (i.e. $d_1 > d_2$). This makes FOGRET easy to recognize as FORGET compared to OFRGET.

(*Figure 1C*) potentially explains why transposing the middle letters renders a word easier to read than transposing its edge letters. This example suggests that orthographic processing can potentially be explained by purely visual processing (as indexed by visual search) without invoking any linguistic factors. However, one must be careful since subjects may have been reading during visual search, thereby activating non-visual lexical or linguistic factors.

To overcome this confound, we asked whether visual search involving letter strings can be explained using a neurally plausible model containing only visual factors. We drew upon two well-established principles of object representations in high-level visual cortex. First, perceptually similar images elicit similar activity in single neurons (*Op de Beeck et al., 2001*; *Sripati and Olson, 2010a*; *Zhivago and Arun, 2014*). Accordingly, we used visual search for single letters to create artificial neurons tuned for letters. Second, the neural response to multiple objects is an average of the individual object responses (*Zoccolan et al., 2005*; *Ghose and Maunsell, 2008*; *Zhivago and Arun, 2014*). Accordingly, we created neural responses to letter strings as a linear sum of single letter responses. We define such responses as *compositional* because the response to wholes is explained by the parts. This stands in contrast to proposals for open bigram detectors (*Grainger and Whitney, 2004*) and for local combination detectors (*Dehaene et al., 2005*; *Dehaene et al., 2010*) according to which reading is enabled by neurons tuned for higher order combinations of letters. Our model only assumes neurons tuned for letter shape and retinal position, as observed in high-level visual cortex (*Lehky and Tanaka, 2016*). It does not capture any information about bigram or higher order detectors, or about other lexical or linguistic factors. We used this model to explain human performance on visual search as well as word recognition tasks. Finally, using brain imaging, we identified the neural substrates for both the letter code as well as subsequent lexical decisions.

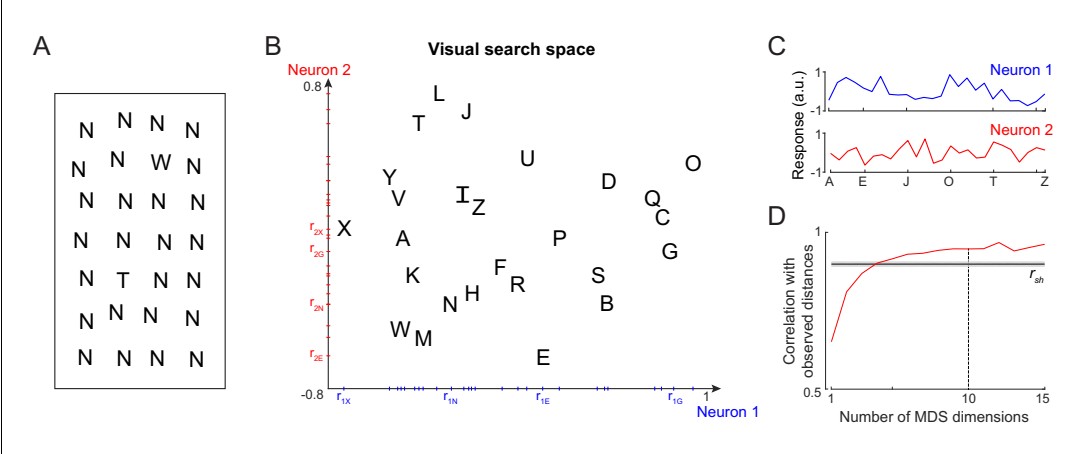

**Figure 2.** Single letter discrimination (Experiment 1). (**A**) Visual search array showing two oddball targets (W and T) among many Ns. It can be seen that finding W is harder compared to finding T. The actual experiment comprised search arrays with only one oddball target among 15 distractors. (**B**) Visual search space for uppercase letters obtained by multidimensional scaling of observed dissimilarities. Nearby letters represent hard searches. Distances in this 2D plot are highly correlated with the observed distances (r = 0.82, p<0.00005). Letter activations along the x-axis are taken as responses of Neuron 1 (*blue*), and along the y-axis are taken as Neuron 2 (*red*), etc. The tick marks indicate the response of each letter along that neuron. (**C**) Responses of Neuron 1 and Neuron 2 shown separately for each letter. Neuron 1 responds best to O, whereas Neuron 2 responds best to L. (**D**) Correlation between observed distances and MDS embedding as a function of number of MDS dimensions. The *black* line represents the split-half correlation with error bars representing s.d calculated across 100 random splits.

## Results

We performed six key experiments and several supporting experiments (reported in the Appendix). In Experiment 1, subjects performed visual search involving single letters, and we used this to construct artificial neurons tuned for letter shape. In Experiments 2–4, we show that search for longer strings can be predicted using these artificial neurons with a simple compositional rule. In Experiment 5, we show that this model also explains human performance on a commonly studied word recognition task. Finally, in Experiment 6, we measured brain activations during word recognition to elucidate the underlying neural representations.

### Experiment 1: Single letter searches

In Experiment 1, subjects had to perform an oddball visual search task involving uppercase letters (n = 26), lowercase letters (n = 26) and digits (n = 10). An example search with two oddball targets is shown in *Figure 2A*, illustrating how finding W is harder compared to finding T in an array of Ns. In the actual experiment, search arrays consisted of only one oddball target among 15 distractors, and subjects had to indicate the side of the screen (let/right) containing the target (see Materials and methods).

Subjects were highly consistent in their responses (split-half correlation between average search times of odd- and even-numbered subjects: r = 0.87, p<0.00005). We calculated the reciprocal of search times for each letter pair which is a measure of distance between them (*Arun, 2012*). These letter dissimilarities were significantly correlated with previously reported subjective dissimilarity ratings (Appendix 1).

Since shape dissimilarity in visual search matches closely with neural dissimilarity in visual cortex (*Sripati and Olson, 2010a*; *Zhivago and Arun, 2014*), we asked whether these letter distances can be used to reconstruct the underlying neural responses to single letters. To do so, we performed a multidimensional scaling (MDS) analysis, which finds the n-dimensional coordinates of all letters such that their distances match the observed visual search distances. In the resulting plot for two dimensions for uppercase letters (*Figure 2B*), nearby letters correspond to small distances that is long search times. The coordinates of letters along a particular dimension can then be taken as the putative response of a single neuron. For example, the first dimension represents the activity of a neuron that responds strongest to the letter O and weakest to X (*Figure 2C*). Likewise the second dimension corresponds to a neuron that responds strongest to L and weakest to E (*Figure 2C*). We note

that the same set of distances can be obtained from a different set of neural responses: a simple coordinate axis rotation would result in another set of neural responses with an equivalent match to the observed distances. Thus, the estimated activity from MDS represents one possible solution to how neurons should respond to individual letters so as to collectively produce behavior.

As expected, increasing the number of MDS dimensions led to increased match to the observed letter dissimilarities (*Figure 2D*). Taking 10 MDS dimensions, which explain nearly 95% of the variance, we obtained the single letter responses of 10 such artificial neurons. We used these single letter responses to predict their response to longer letter strings in all the experiments. Varying this choice yielded qualitatively similar results. Analogous results for all letters and numbers are shown in Appendix 1.

## Experiment 2: Bigram searches

Next, we proceeded to ask whether searches for longer strings can be explained using single letter responses. In Experiment 2, we asked subjects to perform oddball searches involving bigrams. We chose seven uppercase letters (A, D, H, I, M, N, T) and combined them in all possible ways to obtain 49 bigram stimuli. Subjects performed all possible pairs of $^{49}C_2$ searches with one bigram as target and another as distractor (see Materials and methods). An example search is depicted in *Figure 3A*. It can be seen that, finding TA among AT is harder than finding UT among AT. Thus, letter transpositions are more similar compared to letter substitutions, consistent with the classic results on reading (*Norris, 2013*; *Grainger, 2018*). To characterize the effect of bigram frequency, we included both frequent bigrams (e.g. IN, TH) and infrequent bigrams (e.g. MH, HH). As before, subjects were highly consistent in their performance (split-half correlation between odd and even-numbered subjects across all bigrams: r = 0.82, p<0.00005).

Next, we asked whether bigram search performance can be explained using neurons tuned to single letters estimated from Experiment 1. The essential principle for constructing bigram responses is depicted in *Figure 3B*. In monkey visual cortex, the response of single neurons to two simultaneously presented objects is an average of the single object responses (*Zoccolan et al., 2005*; *Zhivago and Arun, 2014*; *Pramod and Arun, 2018*). This averaging can easily be biased through changes in divisive normalization (*Ghose and Maunsell, 2008*). Therefore, we took the response of each neuron to a bigram to be a weighted sum of its responses to the constituent letters (*Figure 3B*). Specifically, the response of a neuron to the bigram AB is given by $r_{AB} = w_1 r_A + w_2 r_B$, where $r_{AB}$ is the response to AB, $r_A$ and $r_B$ are its responses to the constituent letters A and B, and $w_1$, $w_2$ are the summation weights reflecting the importance of letters A and B in the summation. Note that the model also does not incorporate any information specific to a particular bigram and is purely based on combining single letters. Note also that if $w_1 = w_2$, the bigram response to AB and BA will be identical. Thus, discriminating letter transpositions necessarily requires asymmetric summation in at least one of the neurons.

To summarize, the letter model for bigrams has two unknown spatial weighting parameters for each of the 10 neurons, resulting in $2 \times 10 = 20$ free parameters. To calculate dissimilarities between a pair of bigrams, we calculated the Euclidean distance between the 10-dimensional response vectors corresponding to the two bigrams. The data collected in the experiment comprised dissimilarities (1/RT) from 1176 ($^{49}C_2$) searches involving all possible pairs of 49 bigrams. To estimate the model parameters, we optimized them to match the observed bigram dissimilarities using standard nonlinear fitting algorithms (see Materials and methods).

This letter model yielded excellent fits to the observed data (r = 0.85, p<0.00005; *Figure 3C*). To assess whether the model explains all the systematic variance in the data, we calculated an upper bound estimated from the inter-subject consistency (see Materials and methods). This consistency measure ($r_{data} = 0.90$) was close to the model fit, suggesting that the model captured nearly all the systematic variance in the data. As predicted in the schematic figure (*Figure 3B*), the estimated spatial summation weights were unequal (absolute difference between $w_1$ and $w_2$, mean ± sd: 0.07 ± 0.04). To assess whether this difference is statistically significant, we randomly shuffled the observed dissimilarities and estimated these weights. The absolute difference between shuffled weights was significantly smaller than for the original weights (average absolute difference: 0.03 ± 0.02; p<0.005, sign-rank test across 10 neurons).

According to an influential account of word reading, specialized detectors are formed for frequently occurring combinations of letters (*Dehaene et al., 2005*). If this were the case, searches

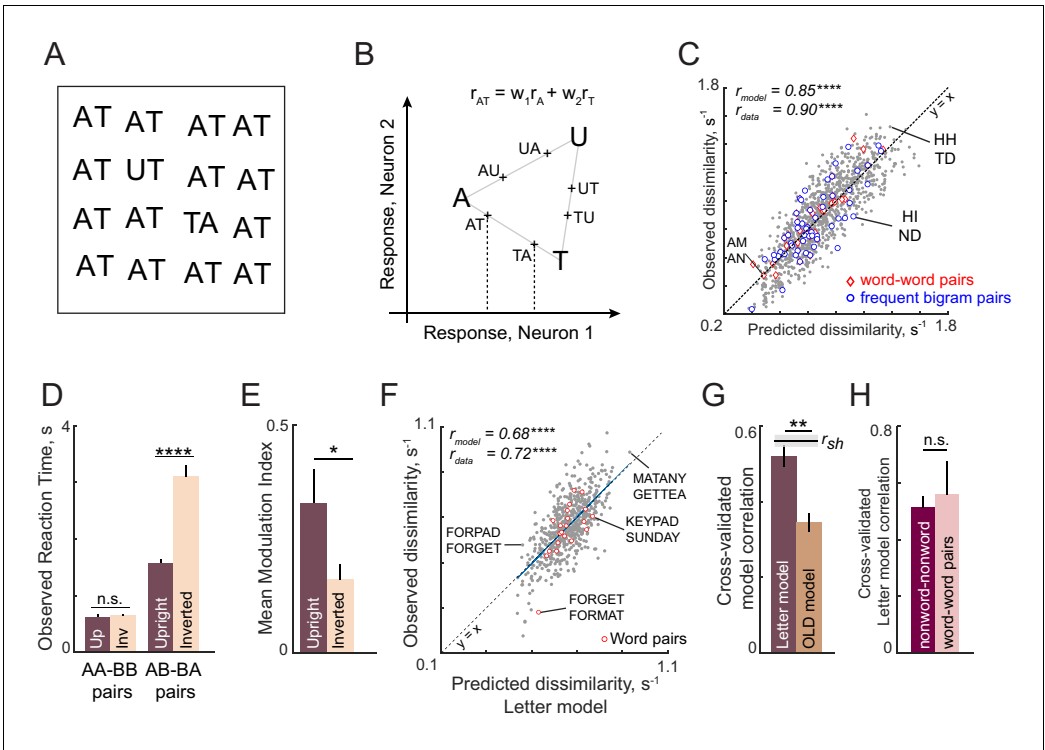

**Figure 3.** Discrimination of strings is explained using single letters (Expts 2–4). (**A**) Example search array with two oddball targets (UT and TA) among the bigram AT. It can be seen that UT is easier to find than TA, showing that letter substitution causes a bigger visual change compared to transposition. (**B**) Schematic diagram of how the bigram response is obtained from letter responses. Consider two neurons selective to single letters A, T and U. These letters can be represented in a 2D space in which the response to each neuron lies along one axis. For each neuron, we take the response to a bigram to be a weighted sum of the single letter responses. Thus, the bigram response lies along the line joining the two stimuli. Note that the bigrams AT and TA can be distinguished only if there is unequal summation. In the schematic, the first position is taken to have higher magnitude, as a result of which the response to AT is closer to A than to T. (**C**) Observed dissimilarities between bigram pairs plotted against predictions of the letter model for word-word pairs (*red diamonds*), frequent bigram pairs (*blue circles*) and all other bigram pairs (*gray dots*), for Experiment 2. Model correlation is shown at the top left, along with the data consistency for comparison. Asterisks indicate the statistical significance of the correlations (**** is $p < 0.00005$). (**D**) Average observed search reaction time for upright (dark) and inverted (pale) bigram searches for repeated letter pairs (AA-BB pairs) and transposed letter pairs (AB-BA pairs) in Experiment 3. Asterisks indicate statistical significance of the main effect of orientation in an ANOVA (see text for details; **** is $p < 0.00005$). (**E**) Mean modulation index of the summation weights, calculated as $|w1-w2|/|w1+w2|$, where w1 and w2 are the bigram summation weights, averaged across the 10 neurons in the letter model for upright (dark) and inverted (pale) bigrams. The asterisk indicates statistical significance calculated on a sign-rank test comparing the modulation index across 10 neurons (* is $p < 0.05$). (**F**) Observed dissimilarities between six-letter strings in visual search (Experiment 4) plotted against predicted dissimilarities from the single letter model for word-word pairs (*red dots*) and all other pairs (*gray dots*). Model correlation is shown at the top left with data consistency for comparison. Asterisks indicate statistical significance of the correlations (**** is $p < 0.00005$). (**G**) Cross-validated model correlation for the letter model (*dark*) and the Orthographic Levenshtein distance (OLD) model (*light*). For each model, the cross-validated correlation is the correlation between model predictions trained on one half of the data and the observed response times from the other half. The upper bound on model fits is the split-half correlation ($r_{sh}$) shown in black with shaded error bars representing standard deviation across 1000 random splits. The asterisk indicates statistical significance of the comparison obtained by estimating the fraction of bootstrap samples in which the observed difference was violated (** is $p < 0.005$). (**H**) Cross-validated letter model correlation for word-word pairs and nonword-nonword pairs.

involving frequent bigrams (e.g. TH, ND) or two letter words (e.g. AN, AM) should produce larger model errors compared to infrequent bigrams, since our model does not incorporate any bigram-selective units. Alternatively, if bigram discrimination was driven entirely by single letters, we should find no difference in errors. In keeping with this latter prediction, we observed no visually obvious difference in model fits for frequent bigram pairs or word-word pairs compared to other bigram pairs (*Figure 3C*). To quantify this observation, we compared the model error (absolute difference between observed and predicted dissimilarity) for the 20 bigram pairs with the largest mean bigram frequency with the model error of the 20 pairs with the lowest mean bigram frequency. This too revealed no systematic difference (mean ± sd of residual error: 0.10 ± 0.08 for the 20 most frequent bigrams and words; 0.11 ± 0.09 for 20 least frequent bigrams; p=0.80, rank-sum test). Thus, model errors are not systematically different for frequent compared to infrequent bigram pairs. We conclude that bigram search can be explained entirely using single neurons tuned to single letters.

## Experiment 3: Upright versus inverted bigrams

In the letter model described above, the response to bigrams is a weighted sum of the single letter responses. As detailed earlier, a critical prediction of this model is that the response to transposed bigrams such as AB and BA will be different only if the summation weights are unequal. By contrast, repeated letter bigrams such as AA and BB will remain discriminable regardless of the nature of summation, since their response will be proportional to the respective single letter responses. Since reading expertise can modulate sensitivity to letter transpositions, we reasoned that familiarity might modulate the summation to make it more asymmetric. We therefore predicted that this would make transposed letter searches (with AB as target and BA as distractor, or vice-versa) easier to discriminate in a familiar upright orientation compared to the (unfamiliar) inverted orientation. By contrast, searches involving repeated letter bigrams (with AA as target and BB as distractor), which also have a change in two letters, will remain equally easy in both upright and inverted orientations.

We tested this prediction in Experiment 3 by asking subjects to perform searches involving upright and inverted bigrams (see Materials and methods). The essential findings are summarized in *Figure 3D*. As predicted, subjects discriminated repeated letter bigrams (AA-BB searches) equally well at both upright and inverted orientations, but were substantially faster at discriminating transposed letter pairs (AB-BA searches) in the upright orientation (*Figure 3D*; for detailed analyses see Appendix 2). We obtained similar results on comparing upright and inverted trigrams as well (Appendix 2). Correspondingly, we observed a larger difference in the model summation weights for upright compared to inverted bigrams (*Figure 3E*).

We conclude that familiarity leads to asymmetric spatial summation. We note, however, that this familiarity could be due to purely visual familiarity of the letters or due to linguistic factors, which we cannot distinguish in our study.

## Experiment 4: Generalization to longer strings

The above analyses show that the letter-based model explains dissimilarities in visual search between bigrams, which rarely contain valid words. We therefore wondered whether these results would extend to longer strings which form words. In Experiment 4, subjects performed visual search involving six-letter strings that were either valid compound words (e.g. FORGET, TEAPOT) or pseudo-words (FORPOT, TEAGET). The single letter model yielded excellent fits to the data (*Figure 3F*). These fits were superior to a widely used measure of string similarity, the Orthographic Levenshtein Distance (OLD) model (*Figure 3G*). Importantly, the letter model fits were equivalent for both word-word pairs and nonword-nonword pairs (*Figure 3H*). These and other analyses are described in Appendix 3.

We performed several experiments to investigate this for other string lengths. Again, the letter model yielded excellent fits across all string lengths (Appendix 4). We also tested lowercase and mixed-case strings because word shape is thought to play a role when letters vary in size or have upward and downward deflections (*Pelli and Tillman, 2007*). Even here, the letter model, without any explicit representation of overall word shape, was able to accurately predict most of the search performance. These results are detailed in Appendix 4.

## Estimating letter dissimilarities from string dissimilarities

The letter model described is neurally plausible and compositional, but is based on dissimilarities between letters presented in isolation. It could be that the representation of a letter within a bigram, although compositional, differs from its representation when seen in isolation. To explore these possibilities we developed an alternate model in which bigram dissimilarities can be predicted using a sum of (unknown) part dissimilarities at different locations. The resulting model, which we denote as the part sum model, yielded comparable fits to the data. It is completely equivalent to the letter model under certain conditions. Unlike the letter model which is nonlinear and could suffer from multiple local minima, the part sum model is linear and its parameters can be estimated uniquely using standard linear regression. Its complexity can be drastically reduced using simplifying assumptions without affecting model fits. These results are detailed in Appendix 5.

## Experiment 5: Lexical decision task

The above experiments show that discrimination of strings in visual search can be explained by neurons tuned for single letter shape with letter responses that combine linearly. Could the same shape representation drive reading behavior? We evaluated this possibility through two separate word recognition experiments.

In Experiment 5, we used a widely used paradigm for word recognition, a lexical decision task (*Norris, 2013*; *Grainger, 2018*), in which subjects have to indicate whether a string of letters is a word or not using a keypress. To develop a quantitative model of lexical decision times, we drew from models of lexical decision in which responses are thought to be based on accumulation of evidence toward or against word status (*Ratcliff et al., 2004*; *Ratcliff and McKoon, 2008*).

Consider what happens when we view the string 'PENICL', as opposed to the string 'EPNCIL' (*Figure 4A*). Since PENICL is visually more similar to the stored word 'PENCIL', it is more likely to be confused with a real word and will take longer to be adjudged a nonword. By contrast, the string 'EPNCIL' will take much less time to respond, since it is far away from any stored word (*Dufau et al., 2012*; *Yap et al., 2015*). Thus, we predict that the response time for a nonword will be inversely proportional to its distance to the nearest word (*Figure 4A*). We also predict that this comparison will be affected by the strength of the stored word representation, such that matches to frequent words are easier. In other words, we predict that response times for nonwords will be inversely proportional to word frequency. Finally, by the same account, when we view the string 'PENCIL', the match to the stored word PENCIL takes no time (the distance being negligible) and the response is therefore dominated by word frequency. We tested these two predictions on the observed lexical decision times.

In this experiment, the words comprised four, five or six-letter words and the nonwords consisted of random strings and jumbled versions of the words (see Materials and methods). Subjects were highly accurate in responding to both words and nonwords (mean ± sd: 96 ± 2% for words, 95 ± 3% for nonwords). Importantly, their response times across words and nonwords were consistent between subjects as evidenced by a significant split-half correlation (correlation between odd- and even-numbered subjects: r = 0.59 for words, r = 0.73 for nonwords, p<0.00005).

We started by characterizing response times for words. To depict the systematic variation in word response times, we plotted them in descending order (*Figure 4B*). Subjects took longer to respond to infrequent words like MALICE compared to frequent words like MUSIC. As predicted, response times for words showed a negative correlation with log word frequency (r = −0.5, p<0.00005 across 450 words). We also estimated other lexical factors such as the logarithm of the letter frequency (averaged across letters of the string), logarithm of the bigram frequency (averaged across all bigrams in the string), and the number of orthographic neighbors (i.e. number of nearby words in the lexicon), which are standard measures in linguistic corpora (see Materials and methods).

To avoid overfitting, we trained a model based on each factor on one half of the subjects and tested it on the other half. This cross-validated performance is shown for all lexical factors in *Figure 4C*. It can be seen that the word frequency is the best predictor of word response times (*Figure 4C*). To assess whether all lexical factors together predict word response times any better, we fit a combined model in which the word response times are modeled as a linear sum of the four factors. The combined model performance was slightly better than the performance of the word frequency model alone (*Figure 4C*). To assess the statistical significance of these results, we performed

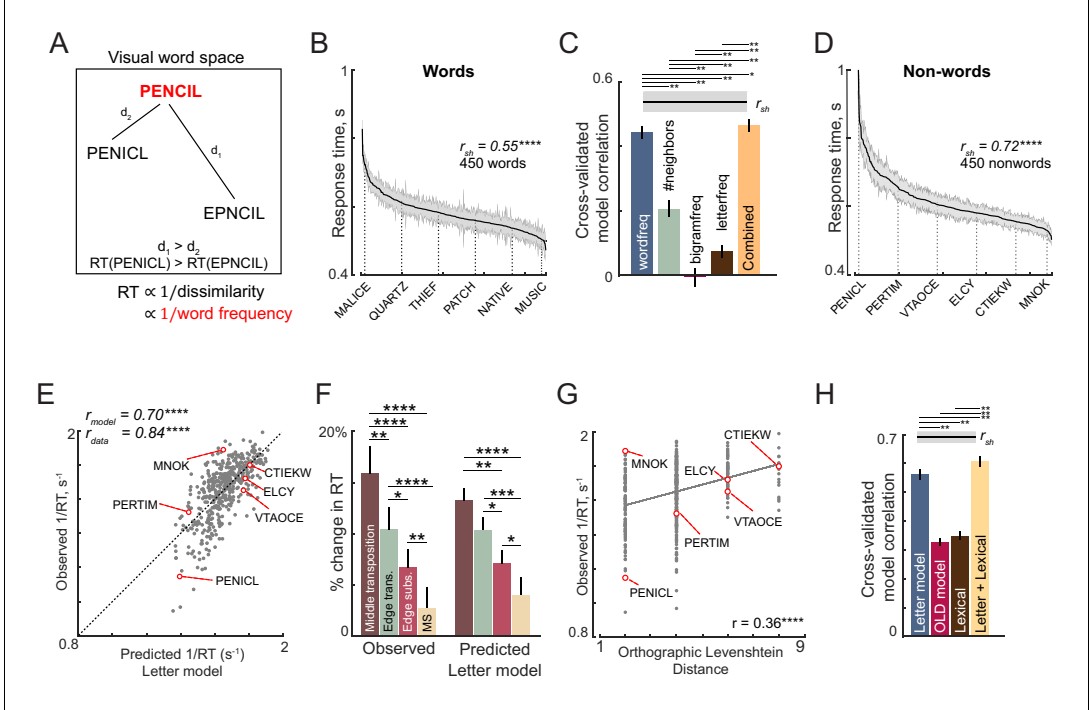

**Figure 4.** Lexical decision task behavior (Experiment 5). (**A**) Schematic of visual word space, with one stored word (PENCIL) and two nonwords (PENICL and EPNCIL). We hypothesize that subjects would take longer to categorize a nonword when it is similar to a word, that is RT for PENICL would be larger than for EPNCIL. Thus, 1/RT would be proportional to this dissimilarity. Likewise we predicted that subjects would be faster to respond to frequent words which have a stronger stored representation. (**B**) Response times for words in the lexical decision task, sorted in descending order. The solid line represents the mean categorization time for words and the shaded bars represent s.e.m. Some example words are indicated using dotted lines. The split-half correlation between subjects ($r_{sh}$) is indicated on the top. (**C**) Cross-validated model correlation between observed and predicted word response times across all words for various models: log word frequency (*blue*), number of orthographic neighbors (*orange*), log mean bigram frequency (*purple*), log mean letter frequency (*cyan*) and a combined model containing all these factors (*red*). Shaded error bars indicate mean ± sd of the correlation across 1000 random splits of the observed data. The asterisk indicates statistical significance of the comparison obtained by estimating the fraction of bootstrap samples in which the observed difference was violated (* is p<0.05, ** is p<0.005). (**D**) Response times for nonwords in the lexical decision task, sorted in descending order. Conventions as in (**A**). (**E**) Observed reciprocal response times for nonwords in the lexical decision task plotted against letter model predictions fit to the full data (450 nonwords). Some example nonwords are depicted. (**F**) Percent change in response time (nonword-RT – word-RT)/word-RT for middle and edge letter transpositions and for middle and edge substitutions for observed data (*left*) and for letter model predictions (*right*). MS: middle substitution. In both cases, asterisks represent statistical significance comparing the means of the corresponding groups using a rank-sum test (* is p<0.05, ** is p<0.005, etc.). (**G**) Observed reciprocal response times plotted against the Orthographic Levenshtein Distance (OLD), a popular model for edit distance between strings. (**H**) Cross-validated model correlation between observed and predicted nonword RTs for the letter model, OLD model, lexical model and the combined neural+lexical model. Conventions are as in (**B**).

a bootstrap analysis. On each trial, we trained all models on the response times obtained from considering only one randomly chosen half of subjects. We calculated the correlation between each model's predictions on the other half of the data, and repeated this procedure 1000 times. Across these samples, the word frequency model performance rarely fell below all other individual models (p<0.005), but was slightly worse than the combined model (p<0.05). We conclude that word response times are determined primarily by word frequency and to a lesser degree by letter frequency. We note that the dependence of word response times on word frequency is non-compositional, since it cannot be explained by letter frequency.

Next we characterized the nonword response times. The nonword responses are plotted in descending order in *Figure 4D*. Subjects took longer to respond to jumbled words like PENICL (original word: PENCIL) with fewer transpositions compared to VTAOCE (original word: OCTAVE) with more transpositions. To test whether nonword to word dissimilarity can predict nonword response times, we took the letter model with 10 neurons (with single letter tuning from visual search) and its spatial summation weights to match the reciprocal of the nonword responses for each word length. We optimized the spatial summation weights based on our observation that summation

weights varied across visual search experiments, and that this could reflect differing attentional resources across letter positions as required for each experiment. This model yielded excellent fits to the data (r = 0.70, p<0.00005; *Figure 4E*) that were comparable to the data consistency ($r_{data}$ = 0.84).

Importantly, this model was able to explain many classic phenomena in orthographic processing. Specifically, subjects took longer to respond to nonwords obtained by transposing a letter of a word, compared to nonwords obtained through letter substitution – these trends were present in the model predictions as well (*Figure 4F*). Likewise, subjects took longer when the middle letters were transposed compared to when the edge letters were transposed – as did the model predictions (*Figure 4F*). These effects replicate the classic orthographic processing effects reported across many studies (*Grainger et al., 2012*; *Norris, 2013*; *Ziegler et al., 2013*; *Grainger, 2018*).

Next we asked whether a widely used measure of orthographic distance could explain the same data. We selected the Orthographic Levenshtein Distance (OLD), in which the net distance between two strings is calculated as the minimum number of letter additions, transpositions and deletions required to transform one string into another. The OLD model yielded relatively poorer predictions of the data (r = 0.36, p<0.00005; *Figure 4G*).

We compared the letter model with two alternate models: the OLD model and a model based on lexical factors. The OLD model is as described above. In the lexical model, the nonword response time is modeled as a linear sum of log word frequency, log mean bigram frequency of words, log mean bigram frequency of nonwords, # orthographic neighbors, log letter frequency. Since all three models have different numbers of free parameters, we compared their performance using cross-validation: we trained each model on one-half of the subjects and evaluated it on the other half of the subjects. The resulting cross-validated model fits are shown in *Figure 4H*. The letter model outperformed both the OLD model and the lexical model (model correlations: r = 0.56 ± 0.02, 0.33 ± 0.01 and 0.35 ± 0.01 for the neural, OLD and lexical models; fraction of bootstrap samples with neural <other models: p<0.005; *Figure 4H*). To be absolutely certain that the superior fit of the letter model was not simply due to having more free parameters, we compared the lexical model with a reduced version of the letter model with only five free parameters (SID model; Appendix 5). Even this reduced model yielded fits were better than the lexical model (SID model correlation: r = 0.48 ± . 02). To assess whether the model trained on visual search data would also be able to predict nonword response times, we took the model trained on the visual search data in Experiment 4, and calculated the word-nonword distances using this model. This too yielded a significant positive correlation (r = 0.39, p<0.00005) that was better than the OLD and lexical models. Finally, a combined model – in which the neural and lexical model predictions were linearly combined – proved to explain more variance than either model (*Figure 4H*).

In sum, we conclude that word response times are explained primarily by word frequency and nonword response times are explained primarily by the distance between the nonword and the nearest word calculated using the compositional neural code.

As a further test of the ability of this compositional code to explain word reading, we performed an additional experiment in which subjects had to recognize the identity of a jumbled word. Here too, response times were explained best by the letter model compared to lexical and OLD models (Appendix 6).

## Experiments 6–7: Neural correlates of lexical decisions

The above results show that visual discrimination of strings can be explained using a letter-based compositional neural code, and that dissimilarities calculated using this code can explain human performance on nonwords during lexical decision tasks. Here, we sought to uncover the brain regions that represent this code and guide eventual lexical decisions. In Experiment 6, we recorded BOLD responses using fMRI while subjects performed a lexical decision task.

Since lexical decision times for nonwords can be predicted using perceptual dissimilarity, we performed a separate experiment to directly estimate perceptual dissimilarities using visual search (Experiment 7; see Materials and methods). Additionally, to compare semantic representations in different ROIs, we estimated the semantic dissimilarity by calculating the cosine distance between GloVe (*Pennington et al., 2014*) feature vectors between word pair (see Materials and methods). Importantly, the perceptual and semantic dissimilarities were uncorrelated (r = 0.03, p=0.55),

thereby allowing us to identify regions with distinct or overlapping perceptual/semantic representations. The perceptual and semantic representations are visualized in Appendix 7.

We identified several possible regions of interest (ROIs) using a combination of functional localizers and anatomical considerations (see Materials and methods). These included the early and mid-level visual areas (V1-V3 and V4), the object-selective lateral occipital region (LO), and two language areas: the visual word form area (VWFA) which selectively responds to words and a broad region in the temporal gyrus reading network (TG). Except for VWFA, all other ROIs were bilateral. The inflated brain map of a representative subject with these ROIs is shown in *Figure 5A*.

In the event-related runs, subjects had to make a response on each trial to indicate whether a string displayed on the screen was a word or not. A total of 64 five-letter strings (32 words and 32 nonwords formed using 10 single letters) were shown. Subjects also viewed the 10 single letters, to which they had to make no response. Subjects were highly accurate (mean ±std of accuracy: 94 ±

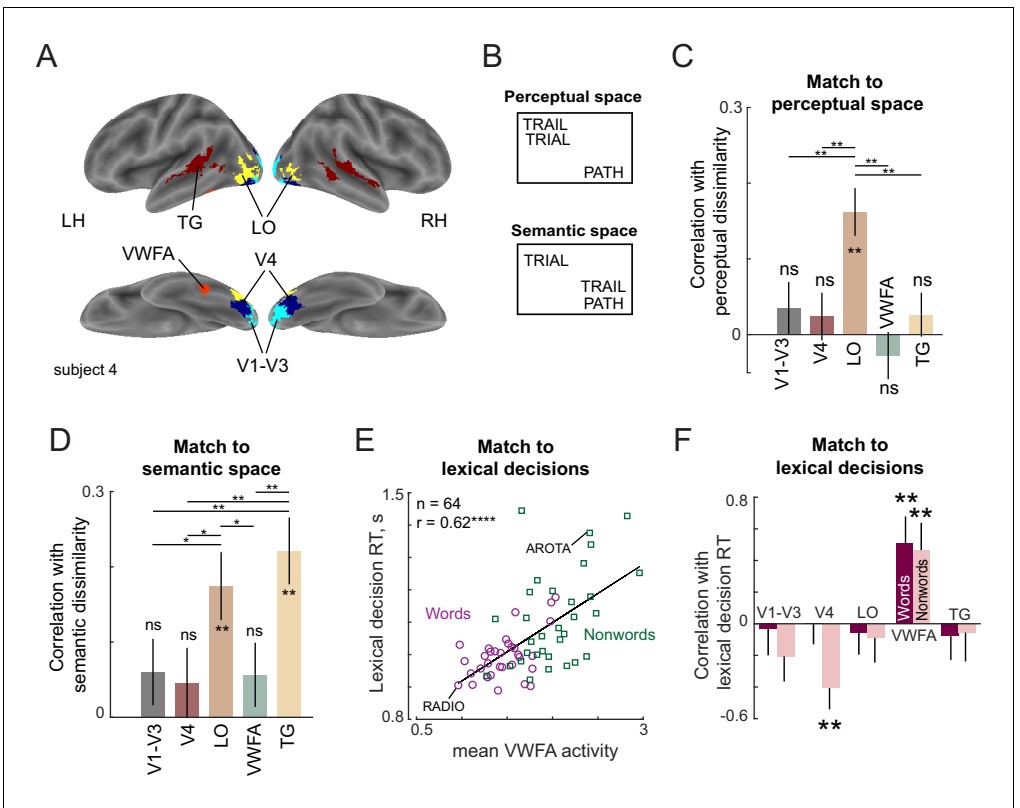

**Figure 5.** Lexical task fMRI (Experiment 6). (**A**) ROIs for an example subject, showing V1–V3 (cyan), V4 (blue), LO (yellow), VWFA (red) and TG (maroon). (**B**) Example difference between perceptual and semantic spaces. In perceptual space, the representation of TRAIL is closer to its visual similar counterpart TRIAL, whereas in semantic space, its representation is closer to its synonym PATH. (**C**) Correlation between neural dissimilarity in each ROI with perceptual dissimilarity between strings measured using visual search (Experiment 7). Error bars indicate standard deviation of the correlation between the group perceptual dissimilarity and ROI dissimilarities calculated repeatedly by resampling of dissimilarity values with replacement across 1000 iterations. Asterisks along the length of each bar indicate statistical significance of the correlation between group behavior and group ROI dissimilarity (** is p<0.005 across 1000 bootstrap samples). Horizontal lines indicate the fraction of bootstrap samples in which the observed difference was violated (* is p<0.05, ** is p<0.005, etc.). All significant comparisons are indicated. (**D**) Correlation between neural dissimilarity in each ROI with semantic dissimilarity for words. Other details are same as in (**C**). (**E**) Correlation between mean VWFA activity (averaged across subjects and voxels) with mean lexical decision time for both words (purple circles) and nonwords (green squares). Each point corresponds to one string and example word and nonword is highlighted. Asterisks indicate statistical significance (**** is p<0.00005). (**F**) Correlation between lexical decision time and mean activity within each ROI separately for words and nonwords. Error bars indicate standard deviation across 1000 bootstrap splits. Asterisks indicate statistical significance (** is p<0.005).

4%) and showed consistent response time variations (split-half correlation between odd and even subjects: $r_{sh}$ = 0.54 and 0.79 for words and nonwords, p<0.00005). As before, the lexical decision time for words was negatively correlated with word frequency (r = −0.42, p<0.05). Likewise, the lexical decision times for nonwords were strongly correlated with the word-nonword dissimilarity measured in visual search in Experiment 7 (r = −0.68, p<0.00005). These results reconfirm the findings of the previous experiment performed outside the scanner.

We then compared the overall brain activation levels for words, nonwords and letters in each ROI. While V4 showed greater activation for words compared to nonwords, VWFA and TG regions showed greater activation to nonwords compared to words, presumably reflecting greater engagement to discriminate nonwords that are highly similar to words (Appendix 7). Although the visual regions did not show differential overall activations, there could still be differential activation at the population level for words and nonwords. This revealed above-chance decoding in all ROIs, and better separation between words and substituted compared to transposed nonwords, matching the trend observed in behavior (Appendix 7).

## Neural basis of perceptual space

Next, we sought to compare the neural representations in each ROI with perceptual and semantic representations. The perceptual and semantic representations can be quite distinct, as depicted in *Figure 5B*: in perceptual space, TRAIL and TRIAL can be quite similar since one is obtained from the other by transposing letters, but the word PATH is distinct. By contrast, in semantic space, TRAIL and PATH have similar meanings and usage whereas TRIAL is distinct. Indeed, perceptual and semantic dissimilarities across words were uncorrelated for the words used in this experiment (r = 0.03, p=0.55).

To investigate these issues, we calculated the neural dissimilarity for each ROI between a given pair of stimuli as the cross-validated Mahalanobis distance between the voxel-wise activations evoked by the two stimuli. We selected this distance metric because it prioritizes the more reliable voxels. The cross-validation procedure calculates Euclidean distances by multiplying activations across runs to avoid bias due to noise. We then averaged this dissimilarity across subjects to get an average neural dissimilarity for that ROI. We then compared this neural dissimilarity in each ROI with perceptual dissimilarities estimated from visual search. This match to perceptual dissimilarity is shown in *Figure 5C*. Among the ROIs tested, only the LO dissimilarities showed a significant correlation (correlation between 1024 pairwise dissimilarities involving $^{32}C_2$ words, $^{32}C_2$ nonwords, and 32 word-nonword pairs: r = 0.16, p<0.00005; *Figure 5C*). A searchlight analysis confirmed that the match to perceptual dissimilarities was strongest in a region centred around the bilateral LO region (Appendix 7). Thus, neural dissimilarity in the LO region match best with the perceptual dissimilarities observed in visual search. We therefore conclude that LO is the likely neural substrate for the compositional letter code.

To further investigate the link between the compositional letter code and the LO representation, we performed several additional analyses. First, we asked whether the neural activation of each voxel in LO could be explained using a linear sum of the single letter activations. Indeed, model fits were comparable for words and nonwords (Appendix 7). This parallels our finding that dissimilarity in visual search was predicted equally well for word-word and nonword-nonword pairs (*Figure 3H*). Second, we confirmed that both the neural tuning for single letters, and the summation weights estimated from the behavioral data in the letter model were qualitatively similar to their counterparts estimated from voxel activations in LO (Appendix 7).

In sum, we conclude that the LO region is the likely neural substrate for the compositional letter code predicted from behavior.

## Neural basis of semantic space

Next we compared neural representations in each ROI to semantic space. The match to semantic space was significant only in the LO and TG regions (correlation between 496 pairwise dissimilarities between words: r = 0.18 ± 0.05 for LO, 0.22 ± 0.04 for TG; *Figure 5D*). A searchlight analysis confirmed that semantic dissimilarities were best correlated with the TG region with additional peaks in prefrontal and motor regions (Appendix 7).

The above analysis shows that neural activations in LO are correlated with both perceptual and semantic dissimilarities, but these correlations cannot be directly compared since they are based on different pairs of stimuli. To investigate whether the neural representation in LO matches better with perceptual or semantic space, we compared the match for word-word pairs alone. This revealed no significant difference between the two correlations (r = 0.16 ± . 04 for LO with visual search, r = 0.16 ± 0.05 for LO with semantic dissimilarites; p=0.49 across 1000 bootstrap samples). To confirm that there is no shared variance between the perceptual and semantic space correlation, we calculated the partial correlation between neural dissimilarities in LO for word-word pairs and the perceptual dissimilarities after factoring out the dependence on semantic dissimilarities (or vice-versa). As expected, both partial correlations were significant (partial correlations: r = 0.13, p<0.005 with perceptual space; r = 0.17, p<0.0005 with semantic space). We conclude that both LO and TG regions represent semantic space.

### Neural basis of lexical decisions

If the LO region represents each string (word or nonword) using a compositional code, then according to the preceding experiments, lexical decisions for words and nonwords must involve some comparison with stored word representations. Recall that lexical decision times for words are correlated with word frequency, and lexical decision times for nonwords are correlated with word-nonword dissimilarity. We therefore asked whether these lexical decision times are correlated with the average activity (across voxels and subjects) in a given ROI. The resulting correlations are shown in *Figure 5F*. Across the ROIs, only the VWFA showed a consistently positive correlation with lexical decision times for both words and nonwords (r = 0.52, p<0.005 for words; r = 0.47, p<0.05 for nonwords, *Figure 5E*). A searchlight analysis confirmed that there was indeed a peak in the correlation with lexical decision times centred on the VWFA, with additional peaks in the parietal and frontal regions (Appendix 7). Interestingly, VWFA activations were larger for nonwords compared to words (mean ± std of VWFA activations across subjects: 1.46 ± 0.22 for words, 2.03 ± 0.28 for nonwords; p<0.005, signed-rank test across 17 subject activations). However, activations were similar for transposed nonwords compared to substituted words (mean ±std VWFA activations across subjects: 1.42 ± 0.33 for transposed nonwords, 1.38 ± 0.33 for substituted nonwords; p=0.62, signed-rank test). We conclude that lexical decisions are driven by the VWFA.

## Discussion

Here, we investigated whether jumbled word reading can be explained using a purely visual representation. We have two major findings. First, we show that a compositional neural code explains visual search for string and responses to nonwords during reading tasks including many orthographic processing phenomena. Second, when subjects performed a lexical decision task, neural dissimilarities in the LO region matched best with perceptual dissimilarities, and lexical decision times were correlated with the activation of the visual word form area (VWFA). This suggests that viewing a string of letters activates a compositional neural code in LO that is subsequently matched with stored word representations in the VWFA. Below we discuss these findings in relation to the existing literature.

### Relation to models of reading

Our compositional letter code stands in stark contrast to existing models of reading. Existing models of reading assume explicit encoding of letter position and do not account for letter shape (*Gomez et al., 2008*; *Davis, 2010*; *Norris and Kinoshita, 2012*; *Norris, 2013*). By contrast, our model encodes letter shape explicitly and position implicitly through asymmetric spatial summation. The implicit coding of letter position avoids the complication of counting transpositions (*Yarkoni et al., 2008*; *Yap et al., 2015*). Our model can thus easily be extended to any language by simply estimating letter dissimilarities using visual search and then estimating the unknown summation weights from visual search for longer strings.

Unlike existing models of reading, our compositional letter code is neurally plausible and grounded in well-known principles of object representations. The first principle is that images that elicit similar activity across neurons in high-level visual cortex will appear perceptually similar (*Op de Beeck et al., 2001*; *Sripati and Olson, 2010a*; *Zhivago and Arun, 2014*). This is non-trivial because

it is not necessarily true in lower visual areas or in image pixels (*Ratan Murty and Arun, 2015*). We have turned this principle around to construct artificial neurons whose shape tuning matches visual search. The second principle is that the neural response to multiple objects is typically the average of the individual object responses (*Zoccolan et al., 2005*; *Sripati and Olson, 2010b*) that can be biased toward a weighted sum (*Ghose and Maunsell, 2008*; *Bao and Tsao, 2018*). Finally, we note that our letter code assumes no explicit calculations of letter position in a word, since the neurons in our model only need to be tuned for retinal position. We speculate that these neurons may be tuned not only to retinal position but also to the relative size and position of letters, as observed in high-level visual cortex (*Sripati and Olson, 2010a*; *Vighneshvel and Arun, 2015*).

## Relation to theories of word recognition

We have found that lexical decisions for nonwords are driven by the dissimilarity between the viewed string and the nearest word. This idea is consistent with the well-known Interactive Activation model (*McClelland and Rumelhart, 1981*; *Rumelhart and McClelland, 1982*), where viewing a string activates the nearest word representation. However, the Interactive Activation model does not explain lexical decisions or scrambled word reading, and also does not integrate letter shape and position into a unified code. Our findings are consistent with previous work showing that non-word responses are influenced by the number of orthographic neighbors (*Yap et al., 2015*). Likewise, we found word frequency to be a major factor influencing lexical decisions, in keeping with previous work (*Ratcliff et al., 2004*; *Dufau et al., 2012*; *Yap et al., 2015*). We note also that personal familiarity with words, as opposed to the word frequency estimated from text corpora, might also influence lexical decisions (*Colombo et al., 2006*; *Kuperman and Van Dyke, 2013*). We have gone further to demonstrate a unified letter-based code that integrates letter shape and position, and localized the underlying neural substrates of the letter code to the LO region, and the comparison process to the VWFA. We propose that the compositional shape code provides a quick match to unscramble a word, failing which subjects may initiate more detailed symbolic manipulation.

The success of our letter code challenges the widely held belief that efficient visual processing of letter strings requires higher-order detectors for letter combinations (*Grainger and Whitney, 2004*; *Dehaene et al., 2005*; *Dehaene et al., 2015*; *Grainger, 2018*). The presence of these specialized detectors should have caused larger model errors for valid words and frequent n-grams, but we observed no such trend (*Figure 3*). However, it is possible that there are combination detectors in subsequent stages where multiple letters have to activate single syllables. So what happens to visual letter representations upon expertise with reading? Our comparison of upright and inverted bigrams suggests that reading should increase letter discrimination and increase the asymmetry of spatial summation (*Figure 3D,E*). This is consistent with our recent finding that reading makes words more predictable from letters (*Agrawal et al., 2019*). It is also consistent with differences in letter position effects for symbols and letters (*Chanceaux and Grainger, 2012*; *Scaltritti et al., 2018*). We propose that both processes may be driven by visual exposure: repeated viewing of letters makes them more discriminable (*Mruczek and Sheinberg, 2005*), while viewing letter combinations induces asymmetric spatial weighting or increased separability. Whether these effects require active discrimination such as letter-sound association training or can be induced even by passive viewing will require comparing letter string discrimination under these paradigms.

## Neural basis of word recognition

Our results elucidate the neural representations that guide lexical decision in several ways. First, we found that perceptual dissimilarities between strings, regardless of word/nonword status, matched best with neural representations in the LO region (*Figure 5C*). This is consistent with similar findings using letters (*Agrawal et al., 2019*) and natural objects (*Khaligh-Razavi and Kriegeskorte, 2014*).

Second, we have found that semantic dissimilarities between words matched both with temporal gyrus regions as well as with LO (*Figure 5D*). The former finding is consistent with temporal gyrus regions participating in the reading network (*Friederici and Gierhan, 2013*), while the latter is concordant with other semantic properties such as animacy encoded in LO (*Bracci and Op de Beeck, 2016*; *Proklova et al., 2016*; *Thorat et al., 2019*). Whether these semantic properties are encoded directly by LO or are a consequence of feedback from language/semantic areas can be distinguished using methods with higher temporal resolution such as MEG or intracranial recordings.

Third, our results confirm and extend our understanding of the VWFA. We found a striking correlation between lexical decision times for words as well as nonwords in the VWFA (*Figure 5E*), suggesting that it is involved in comparing the viewed string with stored words. The finding that VWFA activity is positively correlated with word response times (which reflect word frequency as shown in *Figure 4C*) is consistent with previous studies showing that VWFA activity shows weak activity for frequent words (*Kronbichler et al., 2004*; *Vinckier et al., 2007*). The finding that VWFA activity is correlated with nonword response times (which reflect perceptual distance to the corresponding word, as shown in *Figure 5E*), is consistent with observations that VWFA is modulated by orthographic similarity to words (*Vinckier et al., 2007*; *Baeck et al., 2015*). Finally, our finding that VWFA activations were stronger for nonwords compared to words (*Figure 5E*), has also been observed recently (*Bouhali et al., 2019*). While this might seem paradoxical considering its status as a word form area, the higher activity for nonwords is likely due to many of them being perceptually similar to words, making the lexical decision difficult. That VWFA is activated strongly for hard lexical decisions is also concordant with its higher activation for inverted compared to upright words while making lexical decisions (*Carlos et al., 2019*).

Fourth, our results point a way to resolve contradictory findings regarding VWFA in the literature. Some studies have reported equal activity in VWFA for words and nonwords (*Baker et al., 2007*), and others have reported higher activity for word-like stimuli (*Vinckier et al., 2007*; *Glezer et al., 2009*) – but these observations have been made while subjects performed tasks orthogonal to reading. There have been surprisingly few studies of VWFA activations during word processing tasks (*Baeck et al., 2015*; *Sussman et al., 2018*; *Bouhali et al., 2019*; *Carlos et al., 2019*). By comparing brain activations directly with behavioral responses during a lexical decision task, we found an interesting functional dissociation whereby orthographic (perceptual) similarity between strings was encoded not by VWFA but by LO (*Figure 5C*) and lexical decisions were encoded by VWFA and not LO (*Figure 5F*). This finding implies that most orthographic processing phenomena are driven by compositional neural representations in LO, rather than by the VWFA. These findings are consistent with recent intracranial EEG recordings that report a progression from early to late, or letter-level to word-level representations along the ventral occipitotemporal cortex regions (*Thesen et al., 2012*; *Hirshorn et al., 2016*; *Lochy et al., 2018*). We suggest that fine-grained comparisons between brain activations and behavior will elucidate the roles of the many cortical areas involved in reading.

## Does the compositional letter code explain orthographic processing?

Our letter code explains many orthographic processing phenomena reported in the literature. Its integrated representation of both letter shape and position explains both letter transposition and substitution effects and their relative importance (*Figure 4F*). Its asymmetric spatial weighting favoring the first letter (Appendix 3), explains the first-letter advantage observed previously (*Scaltritti et al., 2018*). It also explains why increasing letter spacing can benefit reading in poor readers, presumably because it increases asymmetry in spatial summation (*Zorzi et al., 2012*).

To elucidate how various jumbled versions of a word are represented according to this neural code, we calculated responses of the letter model trained on data from Experiment 4, and visualized the distances using multidimensional scaling (*Figure 6A*). It can be seen transposing the edge letters (OFRGET) results in a bigger change than transposing the middle letters (FOGRET), thus explaining many transposed letter effects (*Norris, 2013*). Likewise, it can be seen that substituting a dissimilar letter (FORXET) leads to a large change compared to substituting a similar letter (FORCET). Replacing G with C in FORGET leads to a smaller change than replacing with X, thus explaining how priming is stronger when similar letters are substituted (*Marcet and Perea, 2017*). Finally, the letter subset FRGT is closer to FORGET than the same letters reversed (TGRF), thereby explaining subset priming (*Grainger and Whitney, 2004*; *Dehaene et al., 2005*).

Finally, as a powerful demonstration of this code, we used it to arbitrarily manipulate reading difficulty along a sentence (*Figure 6B*), or across multiple transpositions and even number substitutions (*Figure 6C*). We propose that this compositional neural code can serve as a powerful baseline for the purely visual shape-based representation triggered by viewing words, thereby enabling the study of higher order linguistic influences on reading processes.

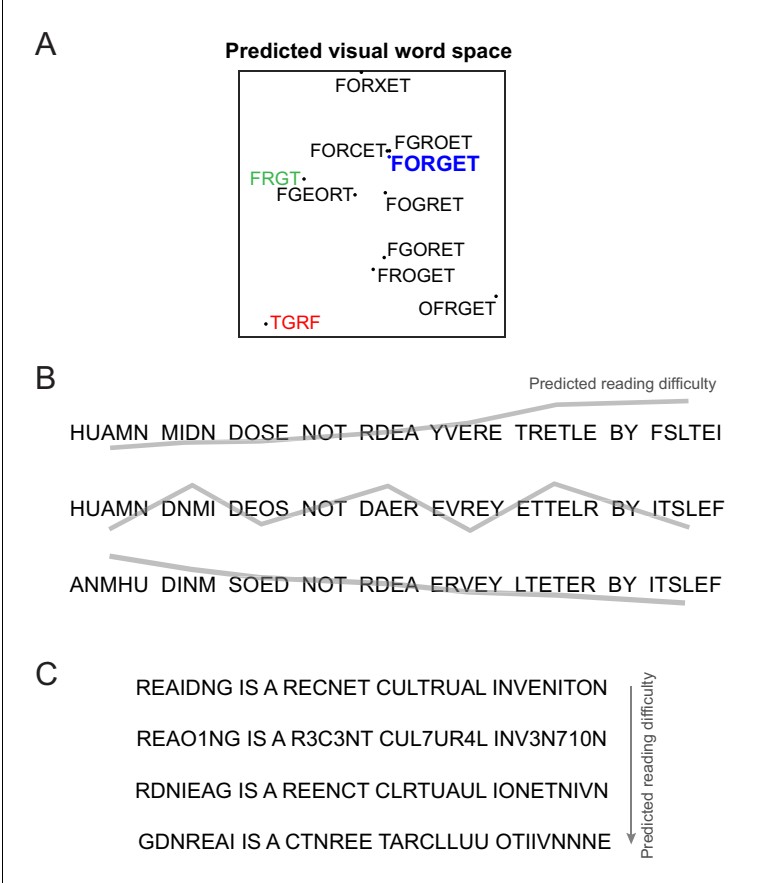

**Figure 6.** Predicting reading difficulty using the letter model. (**A**) Visual word space predicted by the letter model for a word (FORGET) and its jumbled versions. Letter model predictions were based on training the model on compound words (Experiment 4). The plot was obtained by performing multidimensional scaling on the pairwise dissimilarities between strings predicted by the letter model. It can be seen that classic features of orthographic processing are captured by the letter model, including priming effects such as FRGT (*green*) being more similar to FORGET than TGRF (*red*). (**B**) The letter model can be used to sort jumbled words by their reading difficulty, allowing us to create any desired reading difficulty profile along a sentence. *Top row*: Sentence with increasing reading difficulty. *Middle row*: sentence with fluctuating reading difficulty. *Bottom row*: sentence with decreasing reading difficulty. (**C**) The letter model yields a composite measure of reading difficulty that combines letter substitution and transposition effects. Sentences with digit substitutions (*second row*) can thus be placed along a continuum of reading difficulty relative to other sentences (*first, third and fourth rows*) with increasing degree of scrambling.

## Relation between word recognition and reading sentences

Our results constitute an important first step in understanding how we read single words, but reading sentences is much more complex, with potentially many words sampled with each eye movement (*Rayner, 1998*). Our ability to sample multiple letters or words at a single glance is limited by two factors. The first is our visual acuity, which reduces with eccentricity. The second is crowding, by which letters become unrecognizable when flanked by other letters – this effect increases with eccentricity (*Pelli and Tillman, 2008*).

The visual search experiments in our study involved searching for an oddball target (consisting of multiple letters) among multiple distractors. This would most certainly have involved detecting and making saccades to peripheral targets. By contrast, the word recognition tasks in our study involved subjects looking at words presented at the fovea. Our finding that visual search dissimilarity explains word recognition then implies that shape representations are qualitatively similar in the fovea and periphery. Furthermore, the structure of the letter model suggests a possible mechanistic explanation for crowding. Neural responses might show greater sensitivity to spatial location at the fovea

compared to the periphery, leading to more discriminable representations of multiple letters. Alternatively, neural responses to multiple letters might be more predictable from single letters at the fovea but not in the periphery. Both possibilities would predict reduced recognition with closely spaced flankers. Distinguishing these possibilities will require testing neural responses in higher visual areas to single letters and multi-letter strings of both familiar and unfamiliar scripts. Ultimately understanding reading fully will require not only asking how letters combine to form words but also how words combine to form larger units of meaning (*Pallier et al., 2011*; *Nelson et al., 2017*).

## Materials and methods

All subjects had normal or corrected-to-normal vision and gave informed consent to an experimental protocol approved by the Institutional Human Ethics Committee of the Indian Institute of Science (IHEC # 6–15092017). All subjects were fluent English-speaking students at the institute, where English is the medium of instruction. All subjects were multi-lingual and knew at least one other Indian language apart from English.

### Experiment 1 – Single letter searches

#### Procedure

A total of 16 subjects (eight males, 24.4 ± 2.5 years) participated in this experiment. Subjects were seated comfortably in front of a computer monitor placed ~60 cm away under the control of custom programs written in Psychtoolbox (*Brainard, 1997*) and MATLAB. In all experiments, we selected sample sizes based on our previous studies which yielded highly consistent data (*Agrawal et al., 2019*).

#### Stimuli

Single letter images were created using the Arial font. There were 62 stimuli in all comprising 26 uppercase letters (A-Z), 26 lowercase letters (a-z), and 10 digits (0–9). Uppercase stimuli were scaled to have a height of 1°.

#### Task

Subjects were asked to perform an oddball search task without any constraints on eye movements. Each trial began with a fixation cross shown for 0.5 s followed by a 4 × 4 search array (measuring 40° by 25°). The search array always contained only one oddball target with 15 identical distractors. Subject were instructed to locate the oddball target as quickly and as accurately as possible, and respond with a key press ('Z' for left, 'M' for right). A red line divided the screen in two halves. The search display was turned off after the response or after 10 s, whichever was sooner. All stimuli were presented in white against a black background. Incorrect or missed trials were repeated after a random number of other trials. Subjects completed a total of 3782 correct trials ($^{62}C_2$ letter pairs x two repetitions with either letter as target once). For each search pair, the oddball target appeared equally often on the left and right sides so as to avoid creating any response bias. Only correct responses were considered for further analysis. The main experiment was preceded by 20 practice trials involving unrelated stimuli.

#### Data analysis

Subjects were highly accurate on this task (mean ±std: 98 ± 1%). Outliers in the reaction times were removed using built-in routines in MATLAB (*isoutlier* function, MATLAB R2018a). This function removes any value greater than three scaled absolute deviations away from the median, and was applied to each search pair separately. This step removed 6.8% of the response time data, but we obtained qualitatively similar results without this step.

### Estimation of single letter tuning using multidimensional scaling

To estimate neural responses to single letters from the visual search data, we used a multidimensional scaling (MDS) analysis. We first calculated the average search time for each letter pair by averaging across subjects and trials. We then converted this search time (RT) into a distance measure by taking its reciprocal (1/RT). This is a meaningful measure because it represents the underlying rate of

evidence accumulation in visual search (*Sunder and Arun, 2016*), behaves like a mathematical distance metric (*Arun, 2012*) and combines linearly with a variety of factors (*Pramod and Arun, 2014*; *Pramod and Arun, 2016*; *Sunder and Arun, 2016*). Next, we took all pairwise distances between letters and performed MDS to embed letters into n dimensions, where we varied n from 1 to 15. This yielded n-dimensional coordinates corresponding to each letter, whose distances matched best with the observed distances. We then took the activation of each letter along a given dimension as the response of a single neuron. Throughout we performed MDS embedding into 10 dimensions, resulting in single letter responses of 10 neurons. We obtained qualitatively similar results on varying this number of dimensions.

## Estimation of data reliability

To obtain upper bounds on model performance, we reasoned that any model can predict the data as well as the consistency of the data itself. Thus, a model trained on one half of the subjects can only predict the other half as well as the split-half correlation $r_{sh}$. This process was repeated 100 times to obtain the mean and standard deviation of the split-half correlation. However, when a model is trained on all the data, the upper bound will be larger than the split-half correlation. We obtained this upper bound, which represents the reliability of the entire data ($r_{data}$) by applying a Spearman-Brown correction on the split-half correlation, as given by $r_{data} = 2r_{sh}/(r_{sh}+1)$.

## Experiment 2 – Bigram searches

A total of eight subjects (five male, aged 25.6 ± 2.9 years) took part in this experiment. We chose seven uppercase letters (A, D, H, I, M, N, T) and combined them in all possible ways to obtain 49 bigram stimuli. These letters were chosen to maximize the number of two-letter words for example HI, IT, IN, AN, AM, AT, AD, AH, and HA. Letters measured 3° along the longer dimension. Subjects completed 2352 correct trials ($^{49}C_2$ search pairs x two repetitions). All other details were identical to Experiment 1. Letter/Bigram frequencies were obtained from an online database (http://norvig.com/mayzner.html).

### Data analysis

Subjects were highly accurate on this task (mean ±std: 97.6 ± 1.8%). Outliers in the reaction times were removed using built-in routines in MATLAB (*isoutlier* function, MATLAB R2018a). This step removed 8% of the response time data, but we obtained qualitatively similar results without this step.

## Estimating letter model parameters from observed dissimilarities

The total dissimilarity between two bigrams in the letter model is calculated by calculating the average dissimilarity across all neurons. For each neuron, the dissimilarity between bigrams AB and CD is given by:

$$d(AB, CD) = |r_{AB} - r_{CD}| = |(w_1 r_A + w_2 r_B) - (w_1 r_C + w_2 r_D)|$$

where $r_A$, $r_B$, $r_C$ and $r_D$ are the responses of the neuron to individual letters A, B, C and D respectively (derived from single letter dissimilarities), and $w_1$, $w_2$ are the spatial summation weights for the first and second letters of the bigram. Note that $w_1$, $w_2$ are the only free parameters for each neuron.

To estimate the spatial weights of each neuron, we adjusted them so as to minimize the squared error between the observed and predicted dissimilarity. This adjustment was done using standard gradient descent methods starting from randomly initialized weights (*nlinfit* function, MATLAB R2018a). We followed a similar approach for experiments involving longer strings.

## Experiment 3 – Upright and inverted bigrams
### Methods

A total of eight subjects (six males, aged 24 ± 1.5 years) participated in this experiment. Six uppercase letters: A, L, N, R, S, and T were combined in all pairs to form a total of 36 stimuli. These uppercase letters were chosen because their images change when inverted (as opposed to letters like H that are unaffected by inversion), and were chosen to maximize the occurrence of frequent bigrams.

The same stimuli were inverted to create another set of 36 stimuli. Detailed analyses for this experiment are presented in Appendix 2.

### Experiment 4 – compound words

A total of eight subjects (four female, aged 25 ± 2.5 years) participated. Twelve three-letter words were chosen: ANY, FOR, TAR, KEY, SUN, TEA, ONE, MAT, GET, PAD, DAY, POT. Each word was jumbled to obtain 12 three-letter nonwords containing the same letters. The 12 words were combined to form 36 compound words (shown in Appendix 3), such that they appeared equally on the left and right half of the compound words. Detailed analyses for this experiment are included in Appendix 3.

## Calculation of Orthographic Levenshtein Distance (OLD)

For each pair of strings, we calculated the OLD metric using built-in MATLAB function 'editdistance'. This function estimates the number of insertions, deletions, or substitutions are required to convert one string to other. We set the substitution cost to 2, but obtained qualitatively similar results on varying this cost.

### Experiment 5 – Lexical decision task

## Procedure

A total of 16 subjects (nine male, aged 24.8 ± 2.1 years) participated in this task as well as the jumbled word task.

## Stimuli

The stimuli comprised 450 words + 450 nonwords. Words were chosen to avoid multiple possible anagrams (i.e. we avoid words like RATS that could be anagrammed as STAR, ARTS) and to maximize the range of word frequency. The nonwords were either random strings or modified versions of the 450 words (*Table 1*). Strings were presented in uppercase and subtended 1° in visual angle.

## Task

Each trial began a fixation cross shown for 0.75 s followed by a letter string for 0.2 s after which the screen went blank. The trial ended either with the subject's response or after at most 3 s. Subjects were instructed to press 'Z' for words and 'M' for nonwords as quickly and accurately as possible. All stimuli were presented at the centre of the screen and were white letters against a black background. Before starting the main task, subjects were given 20 practice trials using other words and nonwords not included in the main experiment.

## Data analysis

Some nonwords were removed from further analysis due to low accuracy (n = 8, average accuracy <20%). Subjects made accurate responses for both words and nonwords (mean ±std of

**Table 1.** Non-word stimuli in lexical decision task (Experiment 5).

|  | Variations of word ABCDE | four letter words | five letter words | six letter words | Total |
|---|---|---|---|---|---|
| 1) | Edge transpositions: BACDE or ABCED | 15 | 15 | 20 | 50 |
| 2) | Middle transposition: ACBDE or ABDCE | 15 | 15 | 20 | 50 |
| 3) | Two-step edge transposition: CBADE or ABEDC | 0 | 20 | 30 | 50 |
| 4) | Two-step middle transposition: ADCBE | 0 | 20 | 30 | 50 |
| 5) | Random transposition: CDABE, ACDBE, etc. | 25 | 35 | 40 | 100 |
| 6) | Edge substitution: MZCDE or ABCMZ | 15 | 15 | 20 | 50 |
| 7) | Middle substitution: ABMZE | 15 | 15 | 20 | 50 |
| 8) | Random substitution and permutation: MACZE, AMDEZ, etc. | 15 | 15 | 20 | 50 |
|  | Total | 100 | 150 | 200 | 450 |

accuracy: 96 ± 2% for words, 95 ± 3% for nonwords). Outliers in the reaction times were removed using built-in routines in MATLAB (*isoutlier* function, MATLAB R2018a). This step removed 6.4% of the data, but we obtained qualitatively similar results without this step.

## Experiment 6 (Lexical Decision Task – fMRI)

A total of 17 subjects (10 males, 25 ± 4.2 years) participated in this experiment. All subjects were screened for safety and comfort beforehand to avoid adverse outcomes in the scanner.

### Stimuli

The functional localizer block included English words, objects, scrambled words, and scrambled objects. In each run, 14 images were randomly selected from a pool of images. The English words list comprised of 90 five-letter words. Each word was divided into grids of dimension 9 × 3. Scrambled words were generated by randomly shuffling the grids. The object pool comprised 80 naturalistic objects. To generate scrambled objects, the phase of the Fourier transformed images was scrambled and then reconstructed back using inverse Fourier transform. The object images were about 4.5° along the longer dimension and the height of the word stimuli subtended 2° of visual angle.

The event block consisted of 10 single letters and 64 five-letter strings (32 words and 32 nonwords formed using these single letters). The stimulus set comprised of 64 five-letter words and nonwords. The words were chosen from a wide range of frequency of occurrence and the nonwords were created by manipulating the chosen words that is They were: 1) 8-middle transposed version of words, 2) 8-edge transposed version of words, 3) 8-middle substituted version of words, and 4) 8-edge substituted version of words. The stimuli subtended 2° in height, which was the same as in the localizer block. All stimuli were presented as white against a black background.

### Procedure

In the localizer block, a total of 16 images were presented for 0.8 s with an inter stimulus interval of 0.2 s. There were 14 unique stimuli and 2 of them repeated at random time point, in which subjects performed one-back task. Each block ended with a blank screen with fixation cross present for 4 s. Thus, each block lasted 20 s. Each block was repeated thrice in each run.

In the event-related design block, an image was presented at the centre of the screen for 300 ms followed by 3.7 s of blank screen with a fixation cross. In a run, all 74 stimuli were presented once along with 16 trials of fixation cross to jitter inter stimulus interval. Hence there were a total of 92 trials including 4 s fixation trials at the start and end of each run. Each run lasted 376 s. Subjects performed lexical decision task only on strings and were instructed to not press any key for single letters. Overall, subjects completed 2 runs of localizer block, 8 runs of event block and a structural scan block.

### Data acquisition

Subjects viewed images in a mirror-based projection – system. Functional MRI data was acquired using a 32-channel head coil on a 3T Siemens Skyra scanner at HealthCare Global Hospital, Bengaluru. Functional scans were performed using a T2*-weighted gradient-echo-planar imaging sequence with the following parameters: TR = 2 s, TE = 28 ms, flip angle = 79°, voxel size = $3 \times 3 \times 3$ mm$^3$, field of view = $192 \times 192$ mm$^2$, and 33 axial-oblique slices covering the whole brain. Anatomical scans were performed using T1-weighted images with the following parameters: TR = 2.30 s, TE = 1.99 ms, flip angle = 9°, voxel size = $1 \times 1 \times 1$ mm$^3$, field of view = $256 \times 256 \times 176$ mm$^3$.

### Data preprocessing

All raw fMRI data were processed using the SPM 12 toolbox (https://www.fil.ion.ucl.ac.uk/spm/software/spm12/, RRID:SCR_007037). Raw images were realigned, slice-time corrected, co-registered with the anatomical image, segmented, and finally normalized to the MNI305 anatomical template. The results were qualitatively similar without normalization. Smoothing operation was performed only on functional localizer blocks using a Gaussian kernel with FWHM of 5 mm. All SPM parameters were set to default and the voxel size after normalization was set to $3 \times 3 \times 3$ mm$^3$. Prior to normalization, the data was preprocessed using GLMdenoise v1.4 (*Kay et al., 2013*). This step improved

the signal-to-noise ratio in the data by regressing out the noise pattern common across all the voxels in the brain. The noise pattern is estimated from voxels unrelated to the task. The activity corresponding to each condition was estimated by modeling the denoised data using a generalized linear model (GLM) in SPM after removing the low frequency drift using a high-pass filter with a cutoff at 128 s. The event block data was modeled using 89 regressors (74 stimuli + one fixation + six motion regressors + eight runs). The localizer block data was modeled using 13 regressors (four stimuli + one fixation + six motion regressors + two runs).

## ROI definitions

All the regions of interest (ROI) were defined using functional localizer while taking the anatomical location into consideration. Early visual area was defined as the region that responds more to the scrambled object than fixation cross. This functional region was further parsed into V1-V3 and V4 using an anatomical mask from SPM anatomy toolbox (*Eickhoff et al., 2005*). Lateral Occipital (LO) region was defined as a group of voxels that responded more to objects than scrambled objects. The voxels in the LO region was restricted to Inferior Temporal Gyrus, Inferior Occipital Gyrus, and Middle Occipital Gyrus. These anatomical regions were obtained from Tissue Probability Map (TPM) labels in SPM 12. Visual Word Form Area (VWFA) was defined as a region that responded more for words than scrambled words within fusiform Gyrus. The activity for known words was also higher in Superior and Middle Temporal regions. These groups of voxels were grouped under Temporal Gyrus (TG) label. For each contrast, voxel-level threshold of p<0.001 (uncorrected) or cluster level threshold p<0.05 (FWE correction) was used to obtain a contiguous region. For one subject, very few VWFA voxels cross the pre-specified threshold. Hence, the threshold was lowered to p=0.1 (uncorrected). The VWFA voxels were restricted to top-40 voxels (based on T-value in the function localizer contrast). All these regions were visualized on the inflated brain using the BSPMVIEW toolbox (http://www.bobspunt.com/bspmview/).

## Calculation of neural dissimilarity (fMRI)

For each ROI and subject, the pair-wise dissimilarity between any two image pairs was computed using the cross-validated Mahalanobis distance (*rsa.distanceLDC* function, RSA toolbox) (*Nili et al., 2014*). Briefly, it calculates the leave-one-run-out Mahalanobis distance, and the final dissimilarity matrix is estimated by averaging across all the runs. Outliers in dissimilarity values across subjects were removed using built-in routines in MATLAB (*isoutlier* function, MATLAB R2018a). This function was applied to each dissimilarity pair separately, and removed 12.3% of the dissimilarity data. The results were qualitatively similar without this step. The median dissimilarity across all the subjects was considered for further analysis. We obtained qualitatively similar results for other distance measures.

## Calculation of semantic dissimilarity

The semantic distance between every pair of words was computed as the cosine distance between the GloVe feature vectors (*Pennington et al., 2014*) activated by the two words (MATLAB function *word2vec*). These features are based on the co-occurrence statistics of words in a large text corpus, and therefore reflect semantic dissimilarity rather than purely visual dissimilarity.

## Experiment 7 (Five-letter string searches)

A total of 11 subjects (six males, 26 ± 2.7 years) participated in this experiment, of which seven also participated in Experiment 6. Stimuli were identical to Experiment 6, except that they were scaled down to a height of 1° to allow placement in a visual search array. Subjects performed a total of 2048 correct trials ($^{32}C_2$ search pairs x two conditions (words and nonwords) + 32 word-nonword pairs x two repetitions). All trials were interleaved, and incorrect/missed trials appeared randomly later in the task but were not analyzed. All other details were identical to Experiment 1.

## Data analysis

Subjects were highly accurate on this task (mean ±std: 98.6 ± 1%). Outliers in the reaction times were removed using built-in routines in MATLAB (*isoutlier* function, MATLAB R2018a). This step

removed 7% of the response time data, but we obtained qualitatively similar results without this step.

## Acknowledgements

Funding. This research study was funded by Intermediate and Senior Fellowships (Grant Numbers 500027/Z/09/Z and IA/S/17/1/503081 respectively) from the Wellcome Trust/DBT India Alliance and the DBT-IISc partnership program (to SPA).

## Additional information

### Funding

| Funder | Grant reference number | Author |
|---|---|---|
| Wellcome Trust/DBT India Alliance | IA/S/17/1/503081 | SP Arun |
| Wellcome Trust/DBT India Alliance | 500027/Z/09/Z | SP Arun |
| Indian Institute of Science | DBT-IISc partnership programme | SP Arun |

The funders had no role in study design, data collection and interpretation, or the decision to submit the work for publication.

### Author contributions

Aakash Agrawal, Conceptualization, Data curation, Software, Formal analysis, Validation, Investigation, Visualization, Methodology, Writing - original draft, Writing - review and editing; KVS Hari, Conceptualization, Supervision, Writing - review and editing; SP Arun, Conceptualization, Resources, Data curation, Software, Formal analysis, Supervision, Funding acquisition, Validation, Investigation, Visualization, Methodology, Writing - original draft, Project administration, Writing - review and editing

### Author ORCIDs

Aakash Agrawal ⓘ https://orcid.org/0000-0001-8320-4516
KVS Hari ⓘ https://orcid.org/0000-0003-1264-1895
SP Arun ⓘ https://orcid.org/0000-0001-9602-5066

### Ethics

Human subjects: All subjects gave informed consent to an experimental protocol approved by the Institutional Human Ethics Committee of the Indian Institute of Science (IHEC # 6-15092017).

### Decision letter and Author response

Decision letter https://doi.org/10.7554/eLife.54846.sa1
Author response https://doi.org/10.7554/eLife.54846.sa2

## Additional files

### Supplementary files

• Transparent reporting form

### Data availability

Data and code necessary to reproduce the results are available in an Open Science Framework repository at https://doi.org/10.17605/OSF.IO/384ZW.

The following dataset was generated:

| Author(s) | Year | Dataset title | Dataset URL | Database and Identifier |
|---|---|---|---|---|
| VisionLabIISc | 2020 | jumbledwordsfMRI | https://doi.org/10.17605/OSF.IO/384ZW | Open Science Framework, 10.17605/OSF.IO/384ZW |

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

## Appendix 1

### Additional analysis for Experiment 1

The results in the main text were presented for uppercase English letters (*Figure 2*), but in Experiment 1 we also collected visual search data for all English letters and digits (n = 62 characters in all, comprising 26 uppercase + 26 lowercase + 10). We did so in order to predict the visual dissimilarity between letter strings containing both mixed case letters as well as numbers.

To visualize the dissimilarity relations between the 62 characters used, we performed multidimensional scaling. In the resulting plot (*Appendix 1—figure 1A*), nearby characters represent hard searches. A number of interesting patterns can be seen: letters like C, G, Q, O are nearby which is expected given their shared curvatures. Letter pairs such as (M,W) and number pairs such as (6,9) are similar due to mirror confusion (*Vighneshvel and Arun, 2013*).

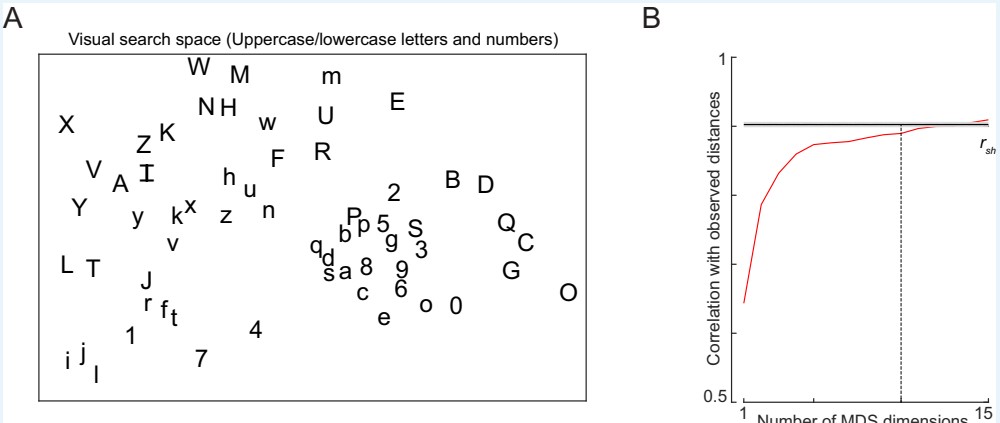

**Appendix 1—figure 1.** Visual search space for letters and digits. (**A**) Visual search space for letters (uppercase and lowercase) and digits obtained by multidimensional scaling of observed dissimilarities. Nearby letters represent hard searches. Distances in this 2D plot are highly correlated with the observed distances (r = 0.79, p<0.00005). (**B**) Correlation between observed distances and MDS embedding as a function of number of MDS dimensions. The horizontal line represents the split-half correlation with error bars representing s.d calculated across 100 random splits.

Next, we investigated the degree to which the observed pairwise dissimilarities are captured by the multidimensional embedding as a function of the number of dimensions. In the resulting plot (*Appendix 1—figure 1B*), it can be seen that nearly 89% of the variance is captured by 10 dimensions as before, which reaches roughly the reliability of the dissimilarity data itself. For the analyses involving mixed case searches or fewer searches, we took a total of six neurons for the letter model, which explain 87.7% of the variance in the pairwise dissimilarities.

### Can letter dissimilarity be predicted using low-level visual features?

To investigate whether single letter dissimilarity can be predicted using low-level visual features, we attempted to predict letter dissimilarities using two models. In the first model, which we call the pixel model, we calculated the dissimilarity between letters to be the absolute difference in pixel intensities between the images of the two letters. This pixel-based model showed a significant correlation (r = 0.50, p<0.00005) but was far from the reliability of the data itself ($r_{sh}$ = 0.90; *Appendix 1—figure 1B*). In the second model, we calculated the

dissimilarity between two letters as the vector distance between the responses evoked by a population of simulated V1 neurons (**Ratan Murty and Arun, 2015**). This V1 model also showed a significant correlation (r = 0.44, p<0.00005) but again far from the reliability of the data itself). We conclude that single letter dissimilarity can only be partially predicted by low-level visual features.

## Is visual search dissimilarity related to subjective dissimilarity?

In this study, we have used visual search as a natural and objective measure for visual dissimilarity. However, previous studies have measured letter dissimilarity either through confusions in letter recognition, or through subjective dissimilarity ratings (**Mueller and Weidemann, 2012**; **Simpson et al., 2013**). We have previously shown that subjective dissimilarity for abstract silhouettes is strongly correlated with visual search dissimilarity (**Pramod and Arun, 2016**). This may not hold for letters since subjects can activate letter representations that are modified through extensive familiarity. To investigate how visual search dissimilarity compares with subjective similarity ratings for letters, we compared search dissimilarities for uppercase letters against two sets of previously reported similarity data. First, we compared visual search dissimilarities with subjective dissimilarity ratings (**Simpson et al., 2013**). This revealed a significant positive correlation (r = 0.69, p<0.0005). Second, we compared visual search dissimilarities with letter confusion data (*3*). To convert letter confusion response times, which are a measure of similarity, into dissimilarities, we took their reciprocals, and then compared them with visual search dissimilarities. This revealed a significant positive, albeit weaker correlation (r = 0.34, p<0.0005).

## Appendix 2

### Upright and inverted bigrams and trigrams

It has been observed that readers are more sensitive to letter transpositions for letters of their familiar script. Since discrimination of letter transpositions in the letter model is a direct consequence of asymmetric spatial summation (main text, *Figure 3*), we predicted that readers should show more asymmetric spatial summation for familiar letters compared to unfamiliar letters. As a strong test of this prediction, we compared visual search performance on upright letters (which are highly familiar) with inverted letters (which are unfamiliar) across two experiments, one on bigrams and the other on trigrams.

The comparison of upright and inverted letter strings is also interesting for a second reason. If reading or familiarity with upright letters led to the formation of specialized detectors for longer strings, then we predict that the letter model (which assumes responses to be driven by single letters only) should yield worse fits for upright compared to inverted letters.

We tested the above two predictions in the following two experiments.

### Experiment 3: Upright vs inverted bigrams

#### Methods

A total of eight subjects (six males, aged 24 ± 1.5 years) participated in this experiment. Six uppercase letters: A, L, N, R, S, and T were combined in all pairs to form a total of 36 stimuli. These uppercase letters were chosen because their images change when inverted (as opposed to letters like H that are unaffected by inversion), and were chosen to maximize the occurrence of frequent bigrams. The same stimuli were inverted to create another set of 36 stimuli. Stimuli subtended ~4° along the longer dimension. Subjects performed all possible searches among the upright letters ($^{36}C_2$ = 630 searches) with two repetitions and likewise for inverted letters. All trials were interleaved. All other details were exactly as in Experiment 2.

#### Results

We observed interesting differences in search difficulty depending on the nature of the bigrams. This pattern is illustrated in *Appendix 2—figure 1A-B*. When the target and distractors consisted of repeated letters (e.g. TT among AA in *Appendix 2—figure 1A*), search is equally easy when the array is upright or inverted. In contrast if the target and distractors are transposed versions of each other (e.g. TA among AT in *Appendix 2—figure 1B*), search is easier in the upright array compared to when it is inverted.

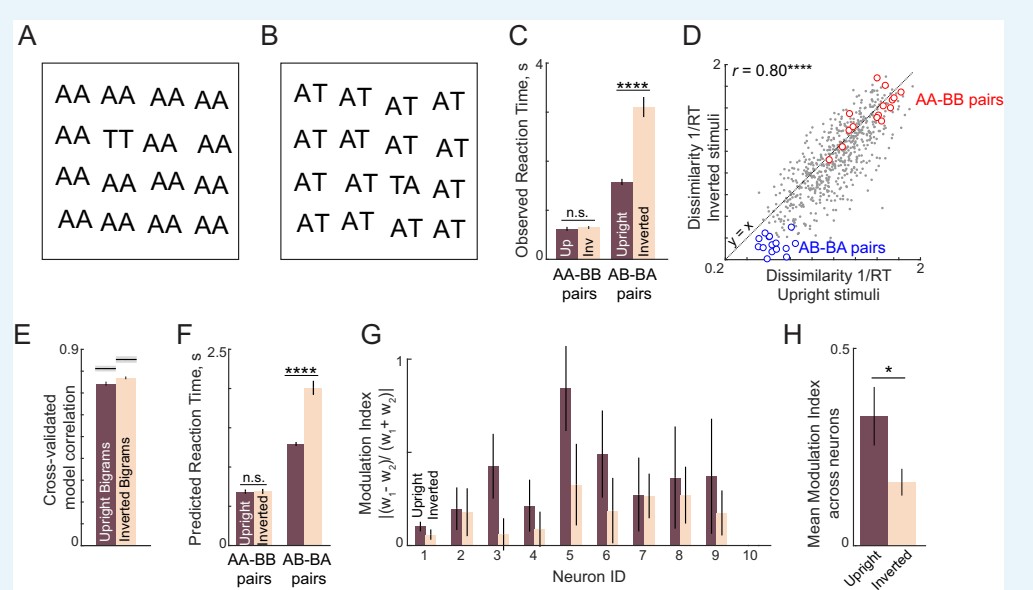

**Appendix 2—figure 1.** Letter model fits for upright and inverted bigrams. (**A**) Example oddball search array for a repeated letter target (TT) among identical repeated-letter distractors (AA). It can be seen that inverting this search array does not affect search difficulty. (**B**) Example oddball search array for transposed letters (TA among AT). It can be seen by inverting this search array makes the search substantially more difficult. (**C**) Average search times in the oddball search task for repeated-letter searches (AA-BB) and transposed letter (AB-BA) searches. Error bars represent s.e.m calculated across subjects. Asterisks represent statistical significance (**** is p<0.00005), as obtained using an ANOVA on the response times with subject, bigram and orientation as factors (see text). (**D**) Dissimilarity of inverted bigram pairs plotted against the dissimilarity of upright bigram pairs. Correlation is shown at the top left. Asterisks indicate statistical significance of the correlations (**** is p<0.00005). (**E**) Cross-validated model correlation of the letter model for upright bigrams and inverted bigrams. *Shaded gray bars* represent the upper bound achievable in each case given the consistency of the data, calculated using the split-half correlation $r_{sh}$. (**F**) Predicted RT from the letter model for repeated letter pairs and transposed letter pairs. Asterisks denote statistical significance as obtained using a sign-rank test on the predicted RTs between upright and inverted conditions. (**G**) Spatial modulation index for each neuron in the letter model for upright and inverted bigrams. (**H**) Average spatial modulation index for upright and inverted bigrams. Asterisks represent statistical significance (* is p<0.05) obtained using a sign-rank test on the spatial modulation index across the 10 neurons.

To confirm that this effect is present across all such pairs, we compared observed response times for these two types of searches between upright and inverted conditions (*Appendix 2—figure 1C*). Response times for the AA-BB searches were comparable for upright and inverted conditions (mean ± sd of RT: 0.66 ± 0.09 s for upright, 0.67 ± 0.1 s for inverted). To assess the statistical significance of this difference, we performed an ANOVA with subject (eight levels), bigram (15 pairs) and orientation (upright vs inverted) as factors. We observed no significant difference in the response times between upright and inverted conditions for AA-BB searches (p=0.65 for main effect of orientation; p<0.00005 for subject and bigram factors, p>0.05 for all interactions).

Next, we compared transposed letter (AB-BA) searches. Here, subjects were clearly faster on the upright searches compared to inverted searches (mean ± sd of RT: 1.58 ± 0.25 s for upright, 3.12 ± 0.76 s for inverted). This difference was statistically significant (p<0.00005 for main effect of orientation; p<0.0005 for subject and p<0.05 for bigram factors, p<0.05 for interactions between pairs and orientation. Other interaction effects were not significant).

To compare bigram dissimilarity between upright and inverted bigrams, we plotted one against the other. This revealed a highly significant correlation (r = 0.80, p<0.00005;

*Appendix 2—figure 1D*). Here too it can be seen that the transposed letter searches are clearly faster when they are upright whereas the repeated letter searches show no such difference.

Thus, inversion slows down transposed letter searches but not repeated letter searches.

## Explaining upright and inverted bigram dissimilarity using the letter model

We fit the letter model to both upright and inverted bigram searches using a total of 10 neurons with single letter responses derived from Experiment 1. The letter model yielded excellent fits on both upright and inverted bigrams. In both cases, the model fits approached the data consistency (*Appendix 2—figure 1E*), implying that the model explained nearly all the explainable variance in the data.

To compare model fits for upright vs inverted bigrams, we performed a bootstrap analysis. Each time, we selected subjects with replacement and fit the letter model to the average dissimilarity computed for this random pool of subjects. Each time, we calculated a normalized correlation measure that takes into account the difference in data reliability between upright and inverted trigram searches. This normalized correlation is simply the model correlation divided by the data consistency. To assess statistical significance, we calculated the fraction of times the normalized correlation in the upright samples was larger than the inverted samples. This analysis revealed significant difference in model performance between upright and inverted searches, but in the opposite direction (average model correlation: r = 0.92 for upright, 0.9 for inverted; fraction of upright <inverted normalized model correlation: p=0). Thus, upright searches are more predictable than inverted searches using the letter model.

Next, we asked whether the letter model can explain the intriguing observation that inversion affects transposed letter searches but not repeated letter searches. This is easy to explain in the letter model: The response to repeated letter bigrams such as AA is unaltered (*Figure 3B*), and therefore the dissimilarity between AA and TT is unaffected by the asymmetry in spatial summation. By contrast, the dissimilarity between transposed letter pairs like AT and TA is directly driven by the asymmetry in spatial summation. We also note that the search TT among AA is much easier than the search for TA among AT. This is also explained by the letter model by the fact that the response to repeated letters is the same as the response to individual letters, leaving their discrimination unaltered. By contrast transposed letters are much more similar since their neural responses are much closer (*Figure 3B*).

To be sure that letter model predictions show the same pattern, we plotted the average response time predicted by the letter model for repeated letter (AA-BB) and transposed letter (AB-BA) searches. To assess the statistical significance, we performed a sign-rank test on the predicted RT. The letter model predictions were exactly as expected (*Appendix 2—figure 1F*).

Next, we analyzed the model parameters in the letter model to ascertain whether the spatial summation in the neurons was indeed different for upright and inverted bigrams. To quantify the degree of asymmetry, we calculated for each neuron a spatial modulation index of the form MI = abs(w1-w2)/(w1+w2) where w1 and w2 are the estimated weights for each letter in the bigram. To avoid unnaturally large modulation indices, w1 and w2 values smaller than 0.01 were set to 0.01. The spatial modulation index for all 10 neurons for upright and inverted bigrams is shown in *Appendix 2—figure 1G*. It can be seen that the modulation index is larger in most cases for the upright bigrams. This difference was statistically significant, as assessed using a sign-rank test on the spatial modulation indices (*Appendix 2—figure 1H*).

## Experiment S1: Upright and inverted trigrams

Here, we asked whether the above results would extend to trigrams. We tested two predictions. First, we predicted greater spatial modulation for upright compared to inverted trigrams, on the premise that better discrimination of trigram transpositions should be driven

by asymmetric spatial summation. Second, if repeated viewing of a trigram or word led to the formation of specialized trigram detectors, then the letter model (which is based only on knowledge of single letters) should produce larger errors compared to other trigrams. We tested this prediction by comparing model fits for searches involving frequent trigrams and words compared to other searches.

## Methods

A total of nine subjects (six females, aged 24.5 ± 2.3 years) participated in the experiment. Six uppercase letters: A, G, N, R, T and Y were combined in all possible three-letter combination to form a total of 216 stimuli. These letters were chosen to include as many three-letter words as possible. In all, 15 three-letter words could be created using these letters (ANT, ANY, ART, GAG, GAY, NAG, NAY, RAG, RAN, RAT, RAY, TAG, TAN, TAR, and TRY).

Since the total number of possible search pairs is large ($^{216}C_2$ = 23,220 pairs), we chose 500 search pairs such that the regression matrix of the part-sum model had full rank that is all the model parameters can be estimated reliably using linear regression. These 500 searches consisted of 368 random search pairs, 105 ($^{15}C_2$) word-word pairs, 15 ($^{3!}C_2$) transposed pairs of nonword comprised of letters G,N, and R. Further, another set of 15 ($^{3!}C_2$) transposed pairs were created using the word TAR. The search pairs formed using the words TAR, ART and RAT were presented only once (although they were counted as both word-word pairs and transposed pairs in the main analysis).

Subjects performed the same searches using upright and inverted trigrams. Stimuli subtended ~5° along the longer dimension. All subjects completed 2000 correct trials (500 searches x 2 orientations x two repetitions). All other details were identical to Experiment 1.

## Results

An example oddball array in the trigram experiment is shown in *Appendix 2—figure 2A*. Note that it is no longer meaningful to compare repeated letter trigrams (AAA-BBB) with transposed trigrams (ABC-BCA) because the repeated letter pairs contain two unique letters, whereas the transposed trigrams contain three unique letters. Subjects were highly consistent in both upright and inverted searches (split-half correlation between even and odd- subjects: r = 0.76 and 0.80, p<0.00005). Upright and inverted dissimilarities were highly correlated (r = 0.80, p<0.00005; *Appendix 2—figure 2B*), although upright searches had higher dissimilarity compared to inverted searches.

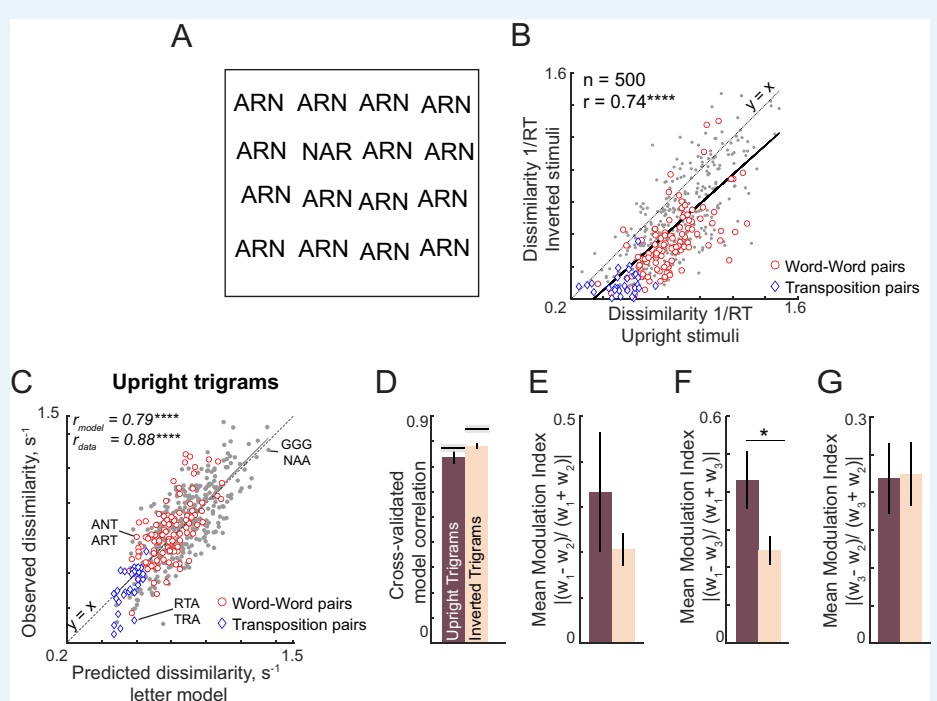

**Appendix 2—figure 2.** Letter model fits for upright and inverted trigrams. (**A**) Example trigram search array containing letter transpositions, with oddball target (NAR) among distractors (ARN). It can be seen that this search is substantially harder when inverted compared to upright. (**B**) Dissimilarity for inverted trigram searches (1/RT) plotted against dissimilarity for upright trigram searches for word-word pairs (red circles, n = 105), transposed letter pairs (blue diamonds, n = 30), and other pairs (gray circles, n = 365). (**C**) Observed dissimilarity for upright trigrams plotted against the predicted dissimilarity from the letter model with symbol conventions as in (**B**). (**D**) Cross-validated letter model correlation for upright and inverted trigrams. (**E**) Average spatial modulation index (across 10 neurons) for the first and second letters in the trigram. (**F**) Same as (**E**) but for the first and third letters. (**G**) Same as (**E**) but for the second and third letters.

Next, we asked whether the letter model can predict dissimilarities between upright trigrams. As before, letter model predictions were highly correlated with the observed data (r = 0.79, p<0.00005; *Appendix 2—figure 2C*) and this model fit approached the data consistency itself ($r_{data}$ = 0.88). Model fits errors were acctually lower for transposed pairs compared to word-word pairs and other pairs (mean ± sd error: 0.1 ± 0.08 for word pairs; 0.07 ± 0.06 for transposed pairs; 0.11 ± 0.08 for other pairs; p=0.02, rank-sum test). The letter model was also able to predict dissimilarities between various trigram transpositions (r = 0.69, p<0.00005; *Appendix 2—figure 2C*). Thus, trigram dissimilarities can be predicted by the letter model regardless of word status or trigram frequency.

We then compared model fits for upright and inverted trigrams. In both cases, the letter model predictions (r = 0.78 and 0.73 for upright and inverted) were close to the consistency of the data ($r_{data}$ = 0.85 and 0.78; *Appendix 2—figure 2D*). To compare these model fits for upright vs inverted statistically, we performed a bootstrap analysis as before (Experiment 3). This analysis revealed no significant difference in model performance between upright and inverted searches (fraction of upright <inverted normalized model correlation: p=0.07).

Finally we asked whether the spatial summation weights of the letter model were systematically different between upright and inverted trigrams. Since there are three spatial modulation weights for each neuron, we calculated the spatial modulation index for all possible pairs of weights (*Appendix 2—figure 2E,F,G*). The spatial modulation ratio was larger for upright compared to inverted trigrams in two of the three pairs, and this difference attained statistical significance for the first and third letters in the trigram (*Appendix 2—*

*figure 2F*). We conclude that the spatial modulation is stronger for upright compared to inverted trigrams.

## Appendix 3

### Additional analysis for Experiment 4 (compound words)

In Experiment 4, we created compound words by combining two valid words such as FORGET from FOR and GET (*Appendix 3—figure 2A*). This resulted in some valid words (e.g. FORGET, TEAPOT) and many invalid words (e.g. FORPOT and TEAGET). The full stimulus set is shown in *Appendix 3—figure 1*.

|  | **ANY** | **FOR** | **TAR** | **KEY** | **SUN** | **TEA** |
|---|---|---|---|---|---|---|
| **ONE** | ANYONE | ONEFOR | ONETAR | KEYONE | ONESUN | TEAONE |
| **MAT** | MATANY | FORMAT | MATTAR | MATKEY | SUNMAT | TEAMAT |
| **GET** | GETANY | FORGET | TARGET | KEYGET | GETSUN | GETTEA |
| **PAD** | PADANY | FORPAD | TARPAD | KEYPAD | PADSUN | PADTEA |
| **DAY** | ANYDAY | DAYFOR | TARDAY | DAYKEY | SUNDAY | DAYTEA |
| **POT** | ANYPOT | POTFOR | POTTAR | POTKEY | SUNPOT | TEAPOT |

**Appendix 3—figure 1.** Stimulus set used for Experiment 4 (Compound Words). The left and the right three letters words were combined to form a six-letter string. The strings that formed compound words are highlighted in red.

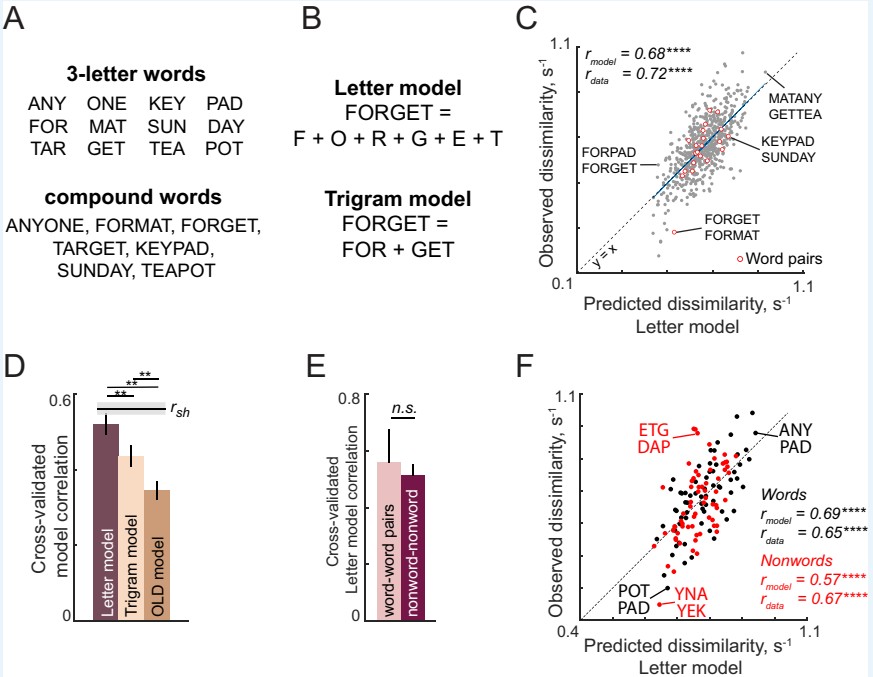

**Appendix 3—figure 2.** Visual search for compound words (Experiment 4). (**A**) Three-letter words (*top*) used to create compound words (*bottom*). (**B**) Illustration of letter and trigram models. In the letter model, the response to a compound word is a weighted sum of responses to the six single letters. In the trigram model, the response to a compound word is a weighted sum of its two trigrams. (**C**) Observed dissimilarity for compound words plotted against predicted dissimilarity from the letter model for word pairs (*red*) and other pairs (*gray*). (**D**) Cross-validated model correlations for the letter model, trigram model and the Orthographic Levenshtein distance (OLD) model. The upper bound on model fits is the split-half correlation ($r_{sh}$), shown in black with shaded error bars representing standard deviation across 30 random splits. Horizontal lines above shaded error bar depicts significant difference

across different models. (**E**) Cross-validated model fits of the letter model for word-words pairs and nonword-nonword pairs. (**F**) Observed dissimilarities for three-letter words (*black*) and nonwords (*red*) plotted against letter model predictions.

If valid words are driven by specialized detectors, responses to valid words should be less predictable by the single letter model. We formulated two specific predictions. First, we hypothesize that the dissimilarity between valid words (e.g. FORMAT vs TEAPOT) would yield larger model errors compared to invalid word pairs (e.g. DAYFOR vs ANYMAT). Second, we predicted that the dissimilarity between two invalid compound words (e.g. DAYFOR vs ANYMAT) should be explained better by their constituent trigrams (DAY, FOR, ANY, MAT) rather than by their constituent letters (*Appendix 3—figure 2B*).

## Methods

A total of eight subjects (four female, aged 25 ± 2.5 years) participated in the experiment. Twelve three-letter words were chosen: ANY, FOR, TAR, KEY, SUN, TEA, ONE, MAT, GET, PAD, DAY, POT. Each word was scrambled to obtain 12 three-letter nonwords containing the same letters. The 12 words were combined to form 36 compound words (*Appendix 3—figure 1*), such that they appeared equally on the left and right half of the compound words. It can be seen that there are seven valid words, whereas the other compound words are pseudowords that carry no meaning. The compound words measured 6° along the longer dimension. Subjects completed 1260 correct trials ($^{36}C_2$ search pairs x two repetitions). Additionally, subjects also performed visual search on three-letter words (n = 132, $^{12}C_2 \times 2$ repetitions) and their jumbled versions (n = 132). Trials timed out after 15 s. All other details were identical to Experiment 1.

Subjects were highly accurate on this task (mean ±std: 98 ± 1%). Outliers in the reaction times were removed using built-in routines in MATLAB (isoutlier function, MATLAB R2018a). This step removed 6.4% of the response time data.

## Results

We recruited eight subjects to perform oddball search involving pairs of trigrams as well as six-letter strings. In all there were 12 three-letter words which resulted in $^{12}C_2$ = 66 searches and 36 compound six-letter strings which resulted in $^{36}C_2$ = 630 searches. We also included 12 three-letter nonwords created by transposing each three-letter words, resulting in an additional $^{12}C_2$ = 66 searches. As before, subjects were highly consistent in their responses (split-half correlation between odd and even subjects: r = 0.54, p<0.00005 for three-letter words; r = 0.46, p<0.00005 for three-letter nonwords; r = 0.65, p<0.00005 for six-letter words).

We started by using the single letter model as before to predict compound word responses. We took single neuron responses as before from Experiment 1, and took the response of each neuron to a compound word to be a weighted sum of its responses to the individual letters. Using these compound word responses, we calculated the dissimilarity between pairs of compound words, and used nonlinear fitting to obtain the best model parameters. The single letter model yielded excellent fits to the data (r = 0.68, p<0.00005; *Appendix 3—figure 2C*). This performance was comparable to the data consistency estimated as before ($r_{data}$ = 0.72).

Next, we asked whether discrimination between compound words can be explained better as a combination of two valid three-letter words, or as a combination of all the constituent six letters. To address this question we constructed a new compositional model based on trigrams, and asked if its performance was better than the single letter model (*Appendix 3—figure 2D*). The trigram-based letter model used trigram dissimilarity to construct neurons with trigram tuning, and spatial summation over the two trigrams to predict the 6-gram responses. To compare the performance of both models even though they have different numbers of free parameters, we used cross-validation: we fit both models on half the subjects and tested their performance on the other half. The letter model outperformed the trigram

model (*Appendix 3—figure 2D*). Because both models were trained on half the subjects and tested on the other half, the upper bound on their performance is simply the split-half correlation between the two halves of the data (denoted by $r_{sh}$). Indeed the letter model performance was close to this upper bound ($r_{sh}$ = 0.56; *Appendix 3—figure 2D*), suggesting that it explained nearly all the explainable variance in the data. Finally, the letter model outperformed a widely used model for orthographic distance – the Orthographic Levenshtein Distance (OLD) (*Appendix 3—figure 2D*). Thus, compound word discrimination can be understood from single letters.

Finally, the letter model fits for word-word pairs and nonword-nonword pairs were not significantly different (*Appendix 3—figure 2E*). This further validates the absence of local combination detectors (*Dehaene et al., 2005*) in perception.

## Three-letter word and nonword dissimilarities

To investigate whether the letter model can predict dissimilarities between three-letter words and non-words, we fit a separate letter model with six neurons as before to the word and non-word dissimilarities. If frequent viewing of words led to the formation of specialized word detectors, the letter model would show worse model fits compared to nonwords. However, we observed no such pattern: the letter model fits were equivalent for words (r = 0.69, p<0.00005; *Appendix 3—figure 2F*) and nonwords (r = 0.57, p<0.00005; *Appendix 3—figure 2F*) – and these fits approached the respective data consistencies ($r_{data}$ = 0.67 for words, 0.68 for nonwords). We conclude that three-letter string dissimilarities can be predicted by the letter model regardless of word status.

## Spatial summation weights

To investigate the spatial summation weights for each neuron, we plotted the estimated spatial summation weights separately (*Appendix 3—figure 3*). It can be seen that spatial summation is heterogeneous across neurons, but the spatial summation of the first neuron follows the characteristic W-shaped curve for letter position observed in studies of reading.

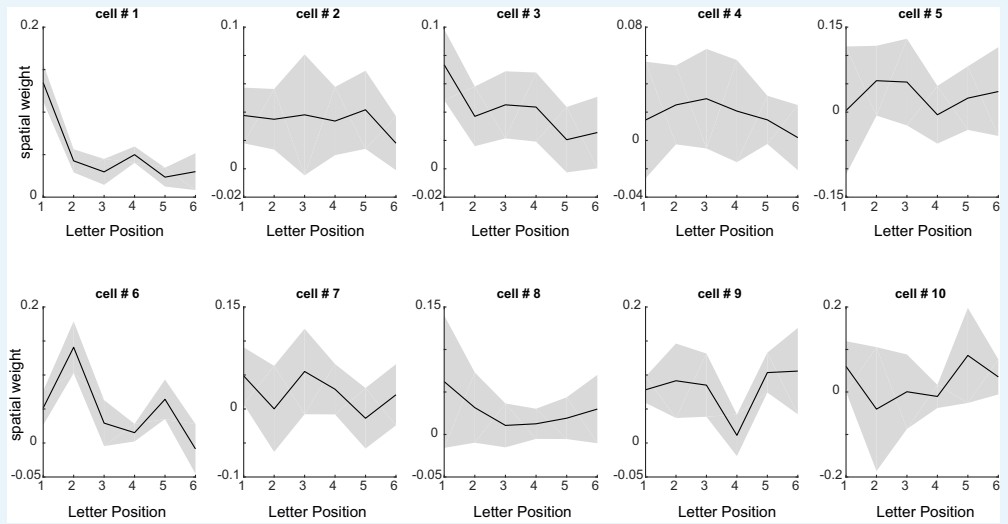

**Appendix 3—figure 3.** Spatial summation weights for each neuron. Estimated spatial summation weights (mean ±std across many random starting points of the nonlinear model fit algorithm) for each neuron in the letter model.

# Appendix 4

## Experiments with longer strings

In the main text, we showed that bigram dissimilarity in visual search can be explained using a simple letter model with single letter responses that match perception, and a compositional spatial summation rule that predicts responses to bigrams. Here we asked whether this approach would generalize to longer strings of letters.

To this end, we performed four additional experiments on longer strings. In Experiment S2, we created trigrams with a fixed middle letter and all possible combinations of flanking letters, to create multiple three-letter words. In Experiment S3, subjects performed searches involving three-, four-, five- and six-letter searches with uppercase, lowercase and mixed case strings. In Experiments S4 and S5, we attempted to optimize the search pairs used to estimate model parameters.

## Methods

### Experiment S2: Trigrams with fixed middle letter

A total of eight subjects (five males, aged $23.9 \pm 1.8$ years) participated in this experiment. Seven uppercase letters: A, E, I, P, S, T and Y were combined (around the stem R that is xRx) in all pairs to form a total of 49 stimuli. These letters were chosen to maximize the occurrence of three-letter words and psuedowords in the stimulus set. The longer dimension of the stimuli was $\sim 5°$. Each subject completed searches corresponding to all possible pairs of stimuli ($^{49}C_2 = 1176$) with two trials for each search. All other details were identical to Experiment 2.

### Experiment S3: Random string searches

A total of 12 subjects (nine female, aged $24.8 \pm 1.64$ years) participated in this experiment. All 26 uppercase and lowercase letters were used to create 1800 stimuli, which were organized into 900 stimulus pairs with varying string length. These 900 pairs comprised 300 6-gram uppercase pairs, 100 6-gram lowercase pairs, 100 6-gram mixed-case pairs, 100 5-gram uppercase pairs, 50 4-gram uppercase pairs, 50 3-gram uppercase pairs and 200 pairs with uppercase strings of differing lengths (50 pairs each of 6- vs 5-grams, 6- vs 4-grams, 5- vs 4-grams, 5- vs 3-grams = 200 pairs total). For each string length, letters were randomly combined to form strings with a constraint that all 26 letters should appear at least once at each location. Each stimulus pair was shown in two searches (with either item as target, and either on the left or right side). The trial timed out at 15 s for all searches.

### Experiment S4 – Optimized four-letter searches

In all, eight subjects (five females, aged $23.5 \pm 2.3$ years) participated in this experiment. To maximize the importance of each spatial location in a four-letter uppercase string, stimuli were created such that there were at least 75 search pairs with the same letter at either of the corresponding locations. Further, to reliably estimate the model parameters, the randomly chosen letters were arranged to minimize the condition number of the linear regression matrix X of the ISI model described below. In all there were 300 search pairs. The trial timed out after 15 s. All other details were similar to Experiment 2.

### Experiment S5 – Optimized six-letter searches

A total of nine subjects (five males, aged $24.1 \pm 2.2$ years) participated in this experiment. We chose 300 search pairs with six-letter strings, according to the same criteria as in Experiment S4. All other details were the same as in Experiment S4.

# Results

Cross-validated model fits across all experiments are shown in **Appendix 4—figure 1**. It can be seen that the letter model fit is close to the split-half consistency of the data. Thus, visual discrimination of longer strings can be explained using a compositional neural code. Below we discuss some experiment-specific findings of interest.

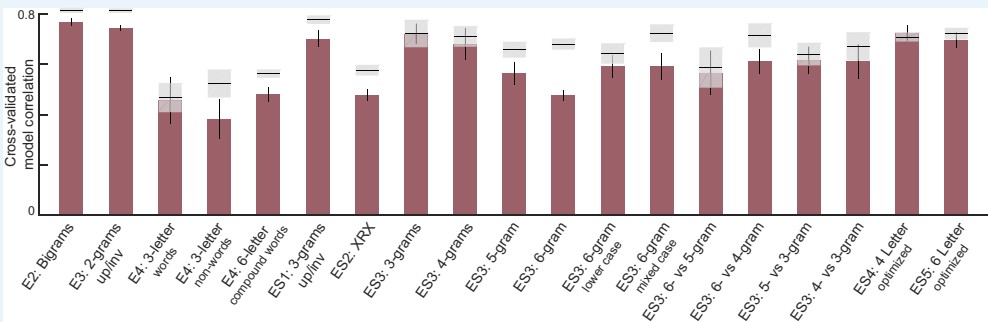

**Appendix 4—figure 1.** Letter model performance for varying length strings. For each experiment, we obtained a cross-validated measure of model performance using six neurons as follows: each time we divided the subjects randomly into two halves, and trained the letter model on one half of the subjects and tested it on the other half. This was repeated for 30 random splits. The correlation between the model predictions and the average dissimilarity from the held-out half of the data was taken to be the model fit. The correlation between the observed dissimilarity between the two random splits of subjects is then the upper bound on model performance (mean ±std shown as *gray shaded bars*).

## Lowercase and mixed-case strings

Word shape is thought to play a role in reading lowercase letters, because of the upward deflection (e.g. l, d) and downward deflections (e.g. p, g) of letters which might confer a specific overall shape to a word. To conclusively establish this would require factoring out the contribution of individual letters to word discrimination, as with the letter model. We were therefore particularly interested in whether the letter model would predict the dissimilarity between lowercase and mixed-case strings where word shape might potentially play a role. As can be seen in **Appendix 4—figure 1**, cross-validated model predictions for lowercase letters were highly correlated with the observed data (r = 0.59, p<0.00005). This correlation approached the upper bound given by the split-half reliability itself ($r_{sh}$ = 0.64). Likewise, model predictions for mixed-case letters were also highly correlated with the observed data (r = 0.59, p<0.00005; **Appendix 4—figure 1**). However, in this case model fits were well below the split-half consistency ($r_{sh}$ = 0.72), suggesting that there is still some systematic unexplained variance in mixed-case strings. This gap in model fit could be simply due to the relatively few mixed-case searches used in this experiment (n = 100), or because of unaccounted factors like word shape. Nonetheless, the letter model explains a substantial fraction of variation in both lowercase and mixed case strings, suggesting that it can be used as a powerful baseline to elucidate the contribution of word shape to reading.

## Unequal length strings

The letter model can be used to calculate responses to any string length, provided the spatial summation weights are known. Given the relatively few searches for unequal lengths in our data, we fit the letter model to unequal length strings using six neurons. Doing so still raised a fundamental issue: which subset of the six spatial summation weights for each neuron should be used to calculate the response to a four-letter string? This requires aligning the four-letter string to the six-letter string in some manner.

To address this issue, we evaluated the letter model fit on four possible alignments between longer and shorter strings, and asked whether model predictions were better for any one alignment compared to others. We aligned the smaller length string to either the left, right, centre or edge of the longer string. Model performance for these different variations is shown in *Appendix 4—table 1*. It can be seen that the model fits are comparable across different choices. However, edge alignment is slightly but not significantly better than other choices. We therefore used edge alignment for all subsequent model predictions.

**Appendix 4—table 1.** Model fits for various choices of string alignment.

| Alignment | Letter model correlation | | | |
| --- | --- | --- | --- | --- |
| | six vs five letter strings | six vs four-letter strings | five vs three-letter strings | four vs three-letter strings |
| Left: ABCDEF vs EFGHxx | 0.54 | 0.66 | 0.58 | 0.57 |
| Right: ABCDEF vs xxEFGH | 0.51 | 0.66 | 0.57 | 0.58 |
| Centre: ABCDEF vs xEFGHx | - | **0.68** | 0.58 | - |
| Edge: ABCDEF vs EFxxGH | **0.55** | 0.63 | **0.60** | **0.59** |

In each case, we fit the letter model with unknown weights corresponding to the longer length. The alignment is indicated by the position of 'x"s in the string. For instance, 'Left' alignment means that a 6-letter string ABCDEF is matched to a four-letter string EFGH by assuming that the response to EFGH is created using the first four weights of spatial summation. Likewise, right alignment means that EFGH is aligned to the right, and therefore its response is created using the last four weights in the six-letter letter model. The best alignment is highlighted for each column in **bold**. None of the correlation coefficient differences were statistically significant ($p > 0.05$, Fisher's z-test).

## Appendix 5

### Estimating letter dissimilarity from bigrams

#### Part-sum model

The letter model described in the text has many desirable features but requires as input the responses to single letters, which were obtained from searches involving single isolated letters. However, it could be that bigram representations can be understood in terms of component letter responses that are different from the responses of letters seen in isolation. It could also be that letter responses are different at each location.

To address these issues, we developed an alternate model in which bigram dissimilarities can be written in terms of unknown single letter dissimilarities. These single letter dissimilarities can be estimated in the model. In this model, which we call the part-sum model, the dissimilarity between two bigrams AB and CD is written as the sum of all pairs of part dissimilarities in the two bigrams (*Appendix 5—figure 1A*). Specifically:

$$d(AB, CD) = CL_{AC} + CR_{BD} + X_{AD} + X_{BC} + W_{AB} + W_{CD} + \text{constant}$$

where $CL_{AC}$ is the dissimilarity between letters at Corresponding Left (CL) locations (A and C), $CR_{BD}$ is the dissimilarity between letters at the Corresponding Right (CR) locations (B and D), $X_{AD}$ and $X_{BC}$ are the dissimilarities between letters across locations in the two bigrams (A and D, B and C), and $W_{AB}$ and $W_{CD}$ are the dissimilarities of letters within each bigram.

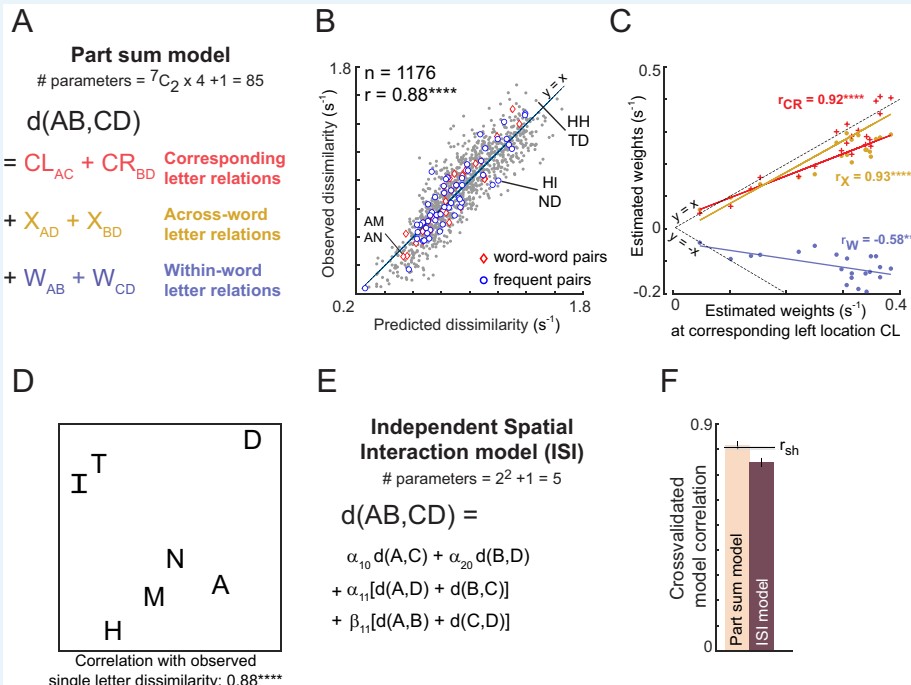

**Appendix 5—figure 1.** Predicting bigram dissimilarity using part-sum model. (**A**) Schematic of the part sum model. According to this model, the dissimilarity (1/RT) between bigrams 'AB' and 'CD' is written as a linear sum of dissimilarities of its corresponding part terms (AC and BD, shown in red), across part terms (AD and BC, shown in yellow), and within part terms (AB and CD, shown in blue). (**B**) Correlation between the observed and predicted dissimilarities (1/ seconds). Each point represents one search pair (n = $^{49}C_2$=1176). Word-word pairs are highlighted using red diamonds, and frequent bigram pairs are highlighted using blue circles. Dotted lines represent unity slope line. (**C**) Correlation between the estimated weights at corresponding location left with estimated weights at 1) corresponding location right (red), 2)

across location (yellow), and 3) within location (blue). Each point represents one letter pair (n = $^7C_2$=21). Dotted lines represent positive and negative unity slope line. (**D**) Perceptual space of the single letter dissimilarities, that are the model coefficients of part terms at left corresponding location (**E**) Schematic of the Independent Spatial Interaction model. In this model, we use the observed letter-pair dissimilarities and only estimate the weights of these letter-pair dissimilarities across different locations. (**F**) Comparing part-sum and ISI model fits. Bar plots represents mean correlation coefficient between the observed and predicted dissimilarities. Error bars represent one standard deviation across 30 splits. Black horizontal line represents mean split-half correlation ($r_{sh}$) and the shaded error bar represents one standard deviation around the mean. (****, p<0.00005, **, p<0.005).

The part-sum model works because a given letter dissimilarity $CL_{AC}$ will occur in the dissimilarity of many bigram pairs (e.g. in the pair AB-CD and in AE-CF) thereby allowing us to estimate its unique contribution. Since there are seven parts, there are $^7C_2$ = 21 possible part-pairs of each type (i.e. for CL, CR, X and W terms), resulting in 21 × 4 = 84 unknown part dissimilarities. Since a given bigram experiment contains all possible $^{49}C_2$ = 1176 bigram searches, there are many more observations than unknowns. The combined set of bigram dissimilarities can be written in the form of a matrix equation **y** = **Xb** where **y** is a 1176 × 1 vector of observed bigram dissimilarities, **X** is a 1176 × 85 matrix containing the number of times (0, 1 or 2) a given letter-pair of each type (CL, CR, X and W) contributes to the overall dissimilarity, and b is a 85 × 1 vector of unknown letter dissimilarities of each type (21 each of CL, CR, X and W and one constant term). The unknown letter dissimilarities of each type was estimated using standard linear regression (*regress* function, MATLAB).

The part sum model has several advantages over the letter model: (1) It is linear which means that its parameters can be uniquely estimated; (2) it is compositional in that the net dissimilarity between two bigrams is explained using the constituent parts without invoking more complex interactions; (3) it can account for potentially different part relations at each location in the two bigrams. We have previously shown that the part-sum model can explain the dissimilarities between a variety of objects (*Pramod and Arun, 2016*).

The part sum model yielded excellent fits to the data (r = 0.88, p<0.00005; *Appendix 5—figure 1B*) that were close to the reliability of the data ($r_{data}$ = 0.90). As before, we observed no systematic deviations between model fits for frequent bigrams compared to infrequent bigrams (*Appendix 5—figure 1B*; average absolute residual error for the top 20 bigram pairs with highest mean bigram frequency: 0.09 ± 0.1 s$^{-1}$; for the bottom-20 bigram pairs: 0.11 ± 0.08 s$^{-1}$; p=0.42, rank-sum test). To assess whether the part dissimilarities of each type (CL, CR, X and W) were related to each other, we plotted each of CR, X and W terms against the CL terms (*Appendix 5—figure 1C*). The CR and X terms were highly positively correlated (*Appendix 5—figure 1C*), whereas the W terms were negative in sign and negatively correlated (*Appendix 5—figure 1C*). The negative values of the W terms means that bigrams with dissimilar letters become less dissimilar, an effect akin to distractor heterogeneity in visual search (*Duncan and Humphreys, 1989*; *Vighneshvel and Arun, 2013*). We conclude that the CL, CR, X and W terms in the part-sum model are driven by a common part representation.

To visualize this underlying letter representation, we performed multidimensional scaling on the estimated part dissimilarities of the CL terms. In the resulting plot, nearby letters represent similar letters (*Appendix 5—figure 1D*). It can be seen that I and T, M and N are similar as in the single-letter representation (*Appendix 1—figure 1A*). These single letter dissimilarities estimated from bigrams using the part-sum model were highly correlated with the single-letter dissimilarities directly observed from visual search with isolated letters (*Appendix 5—figure 1D*).

We conclude that bigram dissimilarities can be predicted from a common underlying letter representation that is identical to that of single isolated letters.

## Equivalence between part-sum and letter model

Given that the part-sum model and letter model both give equivalent fits to the data, we investigated how they are related. Consider a single neuron whose response to a bigram AB is given by: $r_{AB} = \alpha r_A + r_B$, where $r_A$ and $r_B$ are its responses to A & B, and $\alpha$ is the spatial weight

of A relative to B. Similarly its response to the bigram CD can be written as $r_{CD} = \alpha r_C + r_D$. Then the dissimilarity between AB and CD can be written as

$$d(AB, CD)^2$$

$$= (r_{AB} - r_{CD})^2 = (\alpha r_A + r_B - \alpha r_C - r_D)^2$$

$$= \alpha^2(r_A - r_C)^2 + (r_B - r_D)^2 + 2\alpha(r_A - r_C)(r_B - r_D)$$

$$= \alpha^2(r_A - r_C)^2 + (r_B - r_D)^2 + 2\alpha(r_A r_B + r_C r_D - r_A r_D - r_B r_C)$$

$$= \alpha^2(r_A - r_C)^2 + (r_B - r_D)^2 + \alpha\left[(r_A - r_D)^2 + (r_B - r_C)^2 - (r_A - r_B)^2 - (r_C - r_D)^2\right]$$

$$= \alpha^2 d_{AC}^2 + d_{BD}^2 + \alpha\left(d_{AD}^2 + d_{BC}^2\right) - \alpha\left(d_{AB}^2 + d_{CD}^2\right)$$

Thus, the squared dissimilarity between AB and CD can be written as a weighted sum of squared dissimilarities between parts at corresponding locations (A-C and B-D), parts at opposite locations (A-D and B-C) and between parts within each bigram (A-B and C-D), which is essentially the same as the part-sum model. The same argument extends to multiple neurons because the total bigram dissimilarity will be the sum of bigram dissimilarities across all neurons.

There are however two important differences. First, the part sum model is written in terms of a weighted sum of part dissimilarities, whereas the above equation refers to a weighted sum of squared dissimilarities. However, the squared sum of distances and a weighted sum of distances are highly correlated, so the essential relation will still hold. Second, the letter model predicts that the across-bigram terms ($X_{AD}$, $X_{BC}$) should be similar in magnitude but opposite in sign to the within-bigram terms ($W_{AB}$, $W_{CD}$). These weights are similar in magnitude but not exactly equal, as can be seen in Fig S8C. The part-sum model thus allows for greater flexibility in part interactions compared to the letter model.

## Reducing part-sum model complexity (ISI model)

The observation that a common set of letter dissimilarities drive the part-sum model suggests that the part-sum model can be simplified. We therefore devised a reduced version of the part-sum model – called the Independent Spatial Interaction (ISI) model – in which the CL, CR, X and W terms are scaled versions of the single letter dissimilarities (*Appendix 5—figure 1E*). Specifically, the dissimilarity between bigrams AB and CD is:

$$d(AB, CD) = \alpha_{10}d_{AC} + \alpha_{20}d_{BD} + \alpha_{11}(d_{AD} + d_{BC}) + \beta_{11}(d_{AB} + d_{CD}) + c$$

where $d_{AC}$ is the observed dissimilarity between the left letters A & C from visual search and $\alpha_{10}$ is an unknown scaling term, $d_{BD}$ is the observed dissimilarity between the right letters B & D, and $\alpha_{20}$ is an unknown scaling term. Likewise, $\alpha_{11}$ is an unknown scaling term for the net dissimilarity $(d_{AD} + d_{BC})$ between letters across locations, $\beta_{11}$ is the unknown scaling term for the net dissimilarity $(d_{AB} + d_{CD})$ between letters within the two bigrams and c is a constant. Thus, the ISI model has only 5 free parameters: $\alpha_{10}, \alpha_{20}, \alpha_{11}, \beta_{11}$ and c. These parameters can be estimated by solving the matrix equation **y** = **Xb** where **y** is a 1176x1 vector of observed bigram dissimilarities, **X** is a 1176 x 5 matrix containing the net single dissimilarity of each type (CL, CR, X & W) that contributes to the total dissimilarity, and **b** is a 5 x 1 vector of unknown weights corresponding to the contribution of each type of dissimilarity (plus a constant).

The performance of the ISI model is summarized in *Appendix 5—figure 1F*. It can be seen that, despite having only five free parameters compared to 85 parameters of the part-sum model, the ISI model yields comparable fits to the data (*Appendix 5—figure 1F*).

## ISI model performance across all experiments

Next we asked whether the ISI model can be generalized to explain dissimilarities between longer strings. Consider two n-letter strings $u_1 u_2 u_3 u_4 \ldots u_n$ and $v_1 v_2 v_3 v_4 \ldots v_n$. The net dissimilarity between the two strings can be written as:

$$d(u_1 u_2 \ldots u_n, v_1 v_2 \ldots v_n) = \sum_{i=0}^{n} \sum_{k=0}^{n-i} \alpha_{ik}(d(u_i, v_{i+k}) + d(v_i, u_{i+k})) - \sum_{i=0}^{n} \sum_{k=1}^{n-i} \beta_{ik}(d(u_i, u_{i+k}) + d(v_i, v_{i+k})) + c$$

In this manner, we fit the ISI model to all experiments. The resulting cross-validated model fits are shown together with the letter model in *Appendix 5—figure 2*. It can be seen that the ISI model performance is comparable to that of the letter model across all experiments.

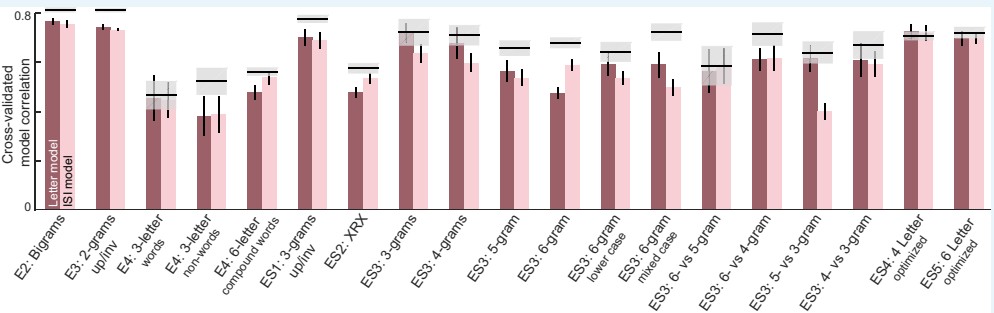

**Appendix 5—figure 2.** ISI and letter model performance across all experiments. For each experiment, we obtained a cross-validated measure of both neural and ISI model performance as follows: each time we divided the subjects randomly into two halves, and trained the letter model on one half of the subjects and tested it on the other half. This was repeated for 30 random splits. The correlation between the model predictions and the average dissimilarity from the held-out half of the data was taken to be the model fit. The correlation between the observed dissimilarity between the two random splits of subjects is then the upper bound on model performance (mean ±std shown as *gray shaded bars*).

## Reducing the complexity of the ISI model

According to the ISI model, the net dissimilarity between two n-grams can be written as a weighted sum of dissimilarities between letter pairs that are varying distances apart. We wondered if the ISI model can be simplified further if there is a systematic pattern whereby these weight corresponding to a given letter pair varies systematically with letter position and distance between the letters.

To assess this possibility, we plotted model coefficients of the ISI model estimated from Experiment S3 along two dimensions. First, we asked if the contribution of letter pairs at corresponding locations in the two n-grams varies with letter position. For varying string lengths (three-, four-, five- and six-letter strings), we observed a characteristic U-shaped function whereby the edge letters contribute more to the net dissimilarity compared to the middle letters (*Appendix 5—figure 3A*). Second, we asked if model weights decrease systematically with inter-letter distance. This was indeed the case regardless of the starting letter in the pair (*Appendix 5—figure 3B*). Finally, we note that across and within part terms are roughly equal in magnitude but opposite in sign (*Appendix 5—figure 1C*).

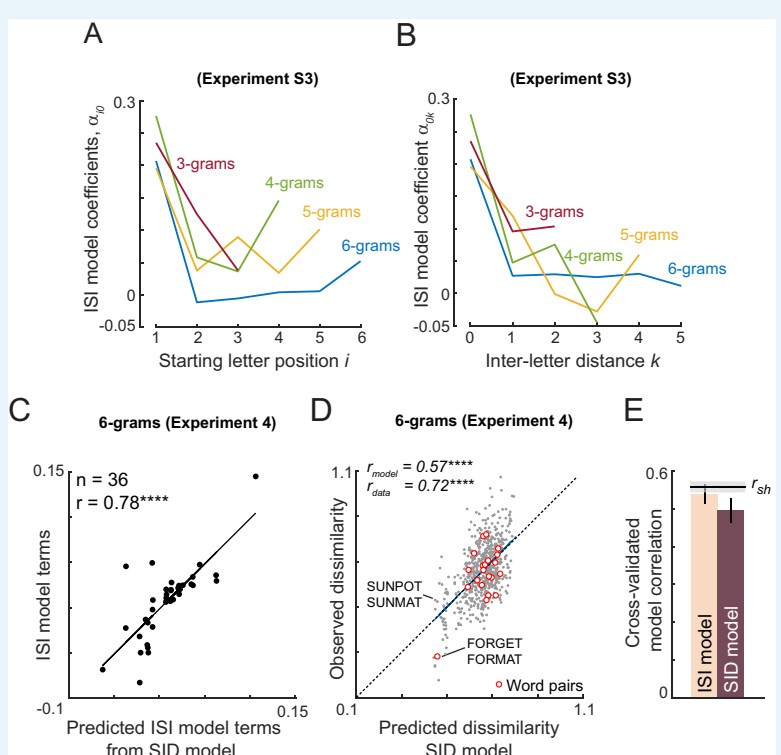

**Appendix 5—figure 3.** Reducing the ISI model. (**A**) ISI model coefficients $\alpha_{i0}$ as a function of starting letter position $i$, for Experiment S3, for varying string lengths. (**B**) ISI model coefficients $\alpha_{1k}$ as a function of inter-letter distance $k$ for Experiment S3, for varying string lengths. (**C**) ISI model coefficients (both $\alpha_{ik}$ and $\beta_{ik}$) plotted against the predicted ISI model coefficients from the SID model. Both models are fitted to data from Experiment 4 (compound words). (**D**) Observed dissimilarity in Experiment four plotted against predicted dissimilarity from the SID model. (**E**) Cross-validated model correlation for ISI and SID models.

The above pattern of weights in the ISI model suggest that we can make two simplifying assumptions. First, the weight of the starting letter is a U-shaped function when the inter-letter distance is zero ($\alpha_{i0}$). Second, weights decrease exponentially thereafter with increasing inter-letter distance. Specifically:

$$\alpha_{i0} = ai^2 + bi + c \ for \ i = 1, 2, ...n$$

$$\alpha_{ik} = \alpha_{i0} e^{-k/\tau} \ for \ k \geq 1$$

$$\beta_{ik} = -\alpha_{ik} \ for \ k \geq 1$$

where a, b, c and τ are the free parameters in this model. This simplified model, which we call the Spatial Interaction Decay (SID) model has only five parameters and can be used to predict the dissimilarities between strings of arbitrary length. The model parameters are obtained using nonlinear gradient descent methods (*nlinfit* function, MATLAB).

To illustrate the performance of the SID model in comparison to the ISI model, we fit the model to 6-letter compound words (Experiment 4). To compare the two models, we plotted the ISI model terms directly estimated from the search data against the ISI model terms predicted from the SID model. This yielded a strong positive correlation (**Appendix 5—figure 3C**). The SID model also yielded excellent fits to the data (**Appendix 5—figure 3D**), and both models yielded comparable fits (**Appendix 5—figure 3E**).

To evaluate this pattern across all experiments, we fit both SID and ISI models to all experiments. Here too we obtained qualitatively similar fits for the two models (**Appendix 5—figure 4**). To confirm whether the SID model trained on one experiment can capture the

variations in another, we trained the SID model on data from Experiment S5 and evaluated it on all other experiments. This too yielded largely similar but smaller predictions (*Appendix 5—figure 4*). This decrease in model fit suggests that model parameters are somewhat dependent on the search pairs chosen.

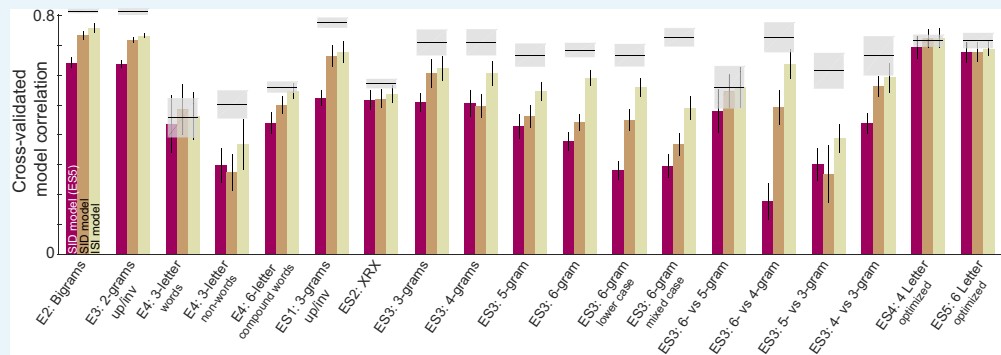

**Appendix 5—figure 4.** ISI and SID model fits across all experiments. Cross-validated model fits for the ISI and SID models across all experiments. In each case the SID and ISI models were fit on a randomly chosen half of the subjects and tested on the other half. The SID (ES5) bars refer to the SID model trained on Experiment S5 and tested on data from a randomly chosen half of subjects in each experiment.

We conclude that dissimilarities between arbitrary letter strings can be predicted using highly simplified models that operate on single letter dissimilarities and simple compositional rules.

## Comparing upright and inverted bigrams using part-sum model

The results in Appendix 2 were based on fitting the letter model to upright and inverted bigrams but assuming a fixed set of single letter responses derived from uppercase letters. The fact that the letter model yielded excellent fits to both upright and inverted bigrams validates this assumption. Nonetheless, we wondered whether differences between upright and inverted bigram searches can be explained solely by different letter representations or by differences in letter interactions.

To investigate this possibility, we fit the part-sum model to upright and inverted bigram searches (*Appendix 5—figure 5A*). The part-sum model also yielded equivalent fits to both upright and inverted searches (*Appendix 5—figure 5B*). If model predictions were similar, we reasoned that the difference between upright and inverted searches must be explained by differences in model parameters. To this end, we compared the estimated letter dissimilarities of each type (CL, CR, X and W) in the upright and inverted searches (*Appendix 5—figure 5C*). Model terms were comparable in magnitude for the CL terms, but were systematically weaker for both CR, X and W terms for inverted compared to upright searches (*Appendix 5—figure 5C*). However in all cases, the recovered letter dissimilarities were correlated between upright and inverted conditions (correlation between upright and inverted model terms: r = 0.93, 0.91, 0.97 and 0.87 for CL, CR, X and W terms; all correlations p<0.00005).

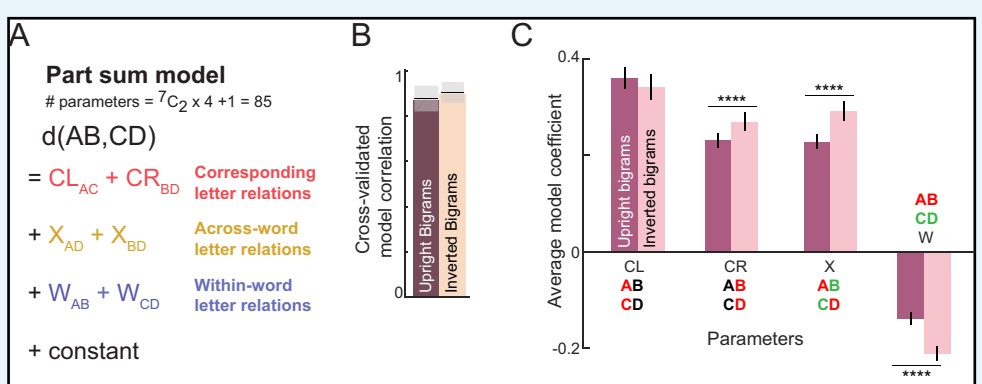

**Appendix 5—figure 5.** Part-sum model fits for upright and inverted bigrams. (**A**) Schematic of the part-sum model, in which the net dissimilarity between two bigrams is given as a linear sum of letter dissimilarities at corresponding locations (CL and CR), across-bigrams (X) and within-bigrams (W). (**B**) Cross-validated model correlation of the part sum model for upright and inverted bigrams. (**C**) Average model coefficients (mean ±sem) of each type for upright and inverted bigrams. Asterisks denote statistical significance (**** is $p<0.00005$) obtained on a sign-rank test comparing 15 letter dissimilarities between upright and inverted conditions).

## Comparing upright and inverted trigrams using part-sum model

The part sum model applied to trigrams is depicted in *Appendix 5—figure 6A*. In this model, the net dissimilarity between two trigrams can be written as a sum of single letter dissimilarities at every possible pair of locations. These locations are grouped as corresponding letters at left (C1), middle (C2) and right (C3) locations, letters across trigrams that are one letter apart starting from the left letter (XN1) or the middle letter (XN2), letters across trigrams that are two letters apart (XF), letters within each trigram that are one letter apart starting from the left letter (WN1) or middle letter (WN2), and letters within each trigram that are two letters apart (WF). Thus the full part-sum model has 9 groups of letter dissimilarities (C1, C2, C3, XN1, XN2, XF, WN1, WN2, WF) each having $^6C_2 = 15$ unknown single letter dissimilarities. Together with a constant term, this part-sum model has $9 \times 15 + 1 = 136$ free parameters. Since we have 500 searches each for upright and inverted trigrams, the part-sum model can be fit to this data to estimate these free parameters using standard linear regression.

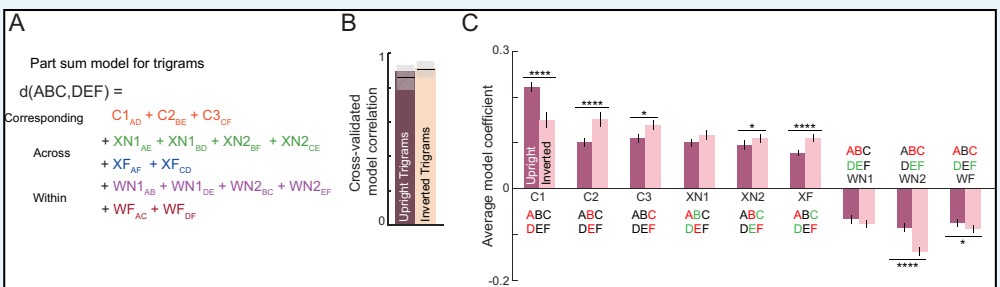

**Appendix 5—figure 6.** Part-sum model fits for upright and inverted trigrams. (**A**) Schematic of part-sum model for trigrams. (**B**) Cross-validated model correlation of part-sum model for upright and inverted trigrams. (**C**) Average model coefficient (averaged across $^6C_2 = 15$ terms) of each type for upright and inverted trigrams. Asterisks indicate statistical significance (* is $p<0.05$, ** is $p<0.005$, etc) calculated using a sign-rank test comparing the upright and inverted model terms. (**D**).

Cross-validated model fits for the part-sum model are shown in *Appendix 5—figure 6B*. It can be seen that the part-sum model explains nearly all the explainable variance in the data for both upright and inverted trigrams (*Appendix 5—figure 6B*). This in turn means that differences between upright and inverted trigrams can be explained using differences in model parameters. This was indeed the case: on plotting the strength of model terms of each type it was clear that seven of the nine types of model terms (C1, C2, C3, XN2, XF, WN2, WF) were systematically larger for upright trigrams compared to inverted trigrams (*Appendix 5—figure 6C*). Finally, we confirmed that model terms for upright and inverted trigrams were highly correlated (correlation between upright and inverted model terms, averaged across nine model term types: $r = 0.65 \pm 0.1$, $p<0.05$ in all cases).

We conclude that upright and inverted trigram searches can be explained using the part-sum model driven by a common single letter representation.

## Jumbled word reading (Experiment S6)

Here, in Experiment S6, we tested subjects on a jumbled word reading task, where they had to view a jumbled word and recognize the original word.

## Methods

### Procedure

A total of 16 subjects (nine male, aged 24.8 ± 2.1 years) participated in the task. Other details were similar to Experiment 5.

### Stimuli

We chose 300 words such that no two words were anagrams of each other. These comprised 75 four-letter words, 150 five-letter words and 75 six-letter words. Jumbled words were created by shuffling two, three, or four letters of each word. There were an equal proportion of two-, three-, and four-letter transpositions. All stimuli were presented in uppercase against a black background.

### Task

Each trial began with a fixation cross shown for 0.5 s followed by a jumbled word that appeared for 5 s (for the first six subjects) and 7 s (for the rest), or until the subject made a response by pressing the space bar on the keyboard. Subjects were asked to press a key as soon as they could recognize the unjumbled word. To ensure that subjects correctly recognized the unjumbled word, they were asked to type the unjumbled word within 10 s of pressing the space bar. The response time was taken as the time at which the subject pressed the space bar. To avoid any memory effects, the same set of jumbled words were shown to all subjects exactly once. We analysed response times only on trials in which the subject subsequently entered the correct word.

### Data analysis

Subjects were reasonably accurate on this task (average accuracy: 59.5 ± 8% across 300 words). Response times for wrongly typed words were discarded. Words correctly solved by more than six subjects (n = 238) were included for further analysis. Since trials were self-paced, we did not remove any outliers in the reaction times. Lexical properties were obtained from the English Lexicon Project (*Balota et al., 2007*).

## Results

Of a total of 300 jumbled words tested, we selected for further analysis 238 words that were correctly unjumbled by more than two-thirds of the subjects. Subjects responded quickly and accurately to these words (mean ±std of accuracy: 71 ± 9%; response time: 2.13 ± 0.33 s across 238 words). Subjects took longer to respond to some jumbled words (e.g. REHID) compared to others (e.g. DBTOU), as seen in the sorted response times (*Appendix 6—figure 1A*). These patterns were consistent across subjects, as evidenced by a significant split-half correlation (r = 0.55, p<0.00005 between odd- and even-numbered subjects).

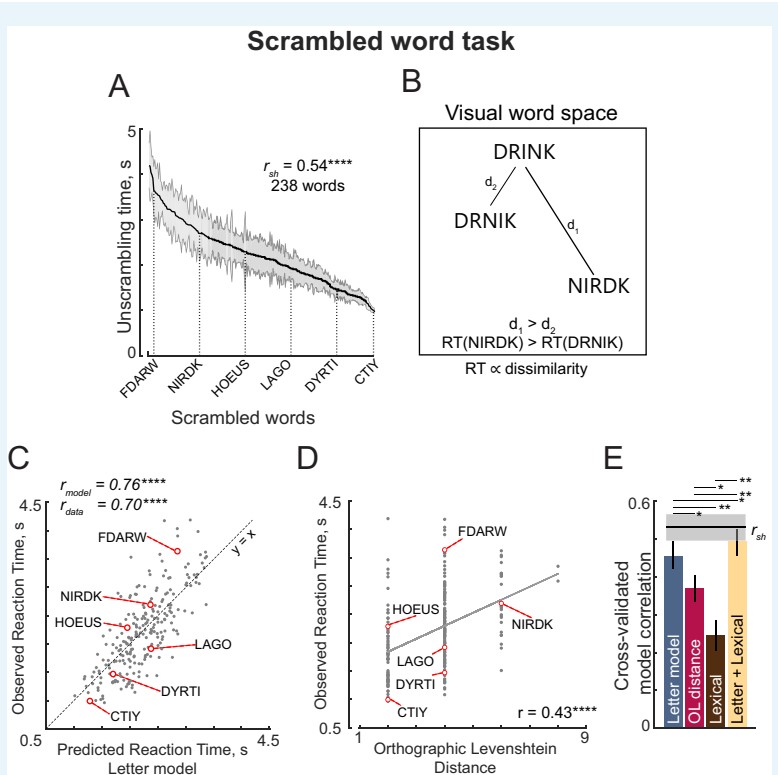

**Appendix 6—figure 1.** Jumbled word task (Experiment S7). (**A**) Response times in the jumbled word task sorted in descending order. Shaded error bars represent s.e.m. Some example words are indicated using dotted lines. The split-half correlation between subjects ($r_{sh}$) is indicated on the top left. (**B**) Schematic of visual word space, with one stored word (DRINK) and two jumbled versions (DRNIK and NIRDK). We predicted that the time taken by subjects to unscramble a jumbled word would be proportional to its dissimilarity to the stored word. Thus, subjects would take longer to unscramble NIRDK compared to DRNIK. (**C**) Observed response times in the jumbled word task plotted against predictions from the letter model based on single letters with spatial summation. Each point represents one word. Asterisks indicate statistical significance (**** is p<0.00005). (**D**) Observed response times in the jumbled word task plotted against Orthographic Levenshtein (OL) distance. Each point represents one word. Asterisks indicate statistical significance (**** is p<0.00005). (**E**) Cross-validated model correlations for the letter model, OLD model, lexical model and the neural +lexical model. Model correlations were obtained by training each model on one half of subjects, and evaluating the correlation on the other half (error bars represent standard deviation across 1000 random splits). The upper bound on model fits is the split-half correlation ($r_{sh}$), shown in black with shaded error bars representing standard deviation across the same random splits. All correlations were individually statistically significant (p<0.00005). Horizontal lines above shaded error bar depicts significant difference across different models that is the fraction of splits in which the observed difference was violated. All significant comparisons are indicated.

Can these patterns in unscrambling time be explained using the letter model? To do so, we reasoned that jumbled words with large dissimilarity to the original word will take longer to elicit a response (***Appendix 6—figure 1B***). Accordingly, we took the average response times to each jumbled word and asked whether it can be predicted using the single letter model described previously. For each word length, we optimized the weights of the single letter model to find the best fit to this data, and then combined the predictions across all word lengths to obtain a composite measure of performance. The single letter model yielded excellent fits to the data (r = 0.76, p<0.00005; ***Appendix 6—figure 1C***). This model fit was comparable to the data consistency ($r_{data}$ = 0.70). An alternate distance model - Orthographic

Levenshtein (OL) distance (*Levenshtein, 1966*) – calculates the number of edits required to transform one string to other. This model neither accounts for letter similarity nor the position of edit. Hence, it fails to account for all the variance in the data (r = 0.44, p<0.00005; *Appendix 6—figure 1D*).

The above finding shows that human performance on unscrambling words is driven primarily by the visual dissimilarity between the jumbled and original word. However, it does not rule out the presence of lexical factors. To assess this possibility, we formulated a model to predict the unscrambling time as a linear sum of many lexical factors. We used five lexical properties: log word frequency, log mean letter frequency, log mean bigram frequency of the jumbled word, log mean bigram frequency of the unjumbled that is original word, and the number of orthographic neighbors (see Materials and methods). To avoid overfitting by either model, we trained both models on one-half of the subjects and tested it on the other half. This lexical model yielded relatively poor fits (r = 0.30, p<0.00005, *Appendix 6—figure 1E*) compared to visual dissimilarity from both single letter model and OL distance model. The difference in model fits was statistically significant (p<0.05, Fisher's z-test). Among the lexical factors, word frequency and letter frequency contributed the most compared to the others (partial correlation of each lexical factor after accounting for all others: r = −0.23, p<0.0005 for log word frequency, r = 0.18, p<0.05 for log mean letter frequency; r = 0.05, p=0.49 for log mean bigram frequency of jumbled word; r = −0.02, p=0.77 for log mean bigram frequency in original word; r = 0.04, p=0.58 for number of orthographic neighbors).

To assess the extent of shared variance in the two models, we calculated the partial correlation between the observed data and the lexical model predictions after factoring out the contribution from visual dissimilarity. This revealed a small partial correlation (r = 0.31, p<0.00005). Conversely, the partial correlation for the single letter model after factoring out the lexical model was much higher (r = 0.75, p<0.00005). Thus, visual dissimilarity from the single letter model dominates jumbled word reading.

Finally, we asked whether both visual dissimilarity and lexical factors contribute to the jumbled word task. We created a combined model in which the jumbled word response times were a linear combination of the predictions of both models. This combined model yielded better predictions than either model by itself (r = 0.78, p<0.00005, *Appendix 6—figure 1E*). To assess the statistical significance of these results, we performed a bootstrap analysis. On each trial, we trained three models on the dissimilarity obtained from considering only one randomly chosen half of subjects: the visual dissimilarity model, the lexical model and the combined model. We calculated the correlation between all three model predictions on the other half of the data, and repeated this procedure 1000 times. The OL distance model does not have any free parameters, hence the distances were directly correlated with the other half of the data. Across these samples, the lexical model fits never exceeded the visual dissimilarity model, suggesting that the visual dissimilarity model was significantly better (p<0.05). Likewise, the combined model was only marginally better than the visual letter model (fraction of combined <visual: p=0.07) but was significantly better than the lexical model (fraction of combined <lexical: p=0).

We conclude that performance on the jumbled word task relies primarily on visual dissimilarity. We propose that this initial visual representation of a word allows the subject to make a quick guess at the correct word without explicit symbolic manipulation.

## Appendix 7

# Additional analyses for Experiments 6 and 7

### Stimulus set

32 words were chosen of varying frequency of occurrence and the nonwords were created by either transposition or substitution of middle or edge letters. 10 single letters: E, S, A, R, O, L, I, T, N, and D were used to form words. The full set of strings used Experiments 6 and 7 is shown below.

**Appendix 7—table 1.** List of 32 words and 32 nonwords used in Experiment 6 & 7. All words and nonwords were created from 10 single letters whose activations were also measured in the experiment.

| Middle Letter Transposition | | Edge Letter Transposition | | Middle Letter Substitution | | Edge Letter Substitution | |
|---|---|---|---|---|---|---|---|
| Words | Nonwords | Words | Nonwords | Words | Nonwords | Words | Nonwords |
| AORTA | AROTA | STOLE | TSOLE | NOISE | NANSE | ONION | ESION |
| DRAIN | DARIN | OASIS | AOSIS | ERROR | EDLOR | RADIO | EEDIO |
| TREND | TERND | SOLID | OSLID | DRILL | DTELL | ASSET | EESET |
| ATLAS | ALTAS | TRAIN | RTAIN | ARISE | AOESE | TEASE | RDASE |
| DRONE | DRNOE | ORDER | ORDRE | LITRE | LINOE | ENTER | ENTRO |
| LEARN | LERAN | INDIA | INDAI | SLIDE | SLONE | IDEAL | IDEDI |
| SANTA | SATNA | RINSE | RINES | NASAL | NATDL | ADORE | ADODI |
| INSET | INEST | SNAIL | SNALI | ALIEN | ALOTN | LASER | LASRO |

## ROI definitions

**Appendix 7—table 2. Variability in ROI definitions across subjects.** For each ROI we report the mean and standard deviation across subjects of the number of voxels, and the XYZ location of the voxel with peak T-value in the normalized brain.

| ROI | Definition | #voxels (mean ± sd) | ROI peak location |
|---|---|---|---|
| V1-V3 | Voxels activated for scrambled > fixation overlaid with anatomical mask of V1-V3 | 398 ± 131 | X: 8 ± 17 Y: -96 ± 5 Z: 6 ± 9 |
| V4 | Voxels activated for scrambled > fixation overlaid with anatomical mask of V4 | 185 ± 63 | X: 5 ± 26 Y: -88 ± 3 Z: 27 ± 11 |
| LO | Voxels activated for object > scrambled and not in other ROIs | 371 ± 115 | X: -17 ± 43 Y: -66 ± 15 Z: -19 ± 5 |
| VWFA | Voxels with known words > scrambled word in a contiguous region in fusiform gyrus | 52 ± 15 | X: -44 ± 4 Y: -50 ± 5 Z: -17 ± 5 |
| TG | Voxels with native words > scrambled word in a contiguous region in temporal gyrus | 289 ± 182 | X: -44 ± 39 Y: -43 ± 18 Z: 3 ± 9 |

# Visualization of perceptual and semantic space

To visualize words and nonwords in perceptual space, we performed a multidimensional scaling (MDS) analysis of the visual search data (Experiment 7). Briefly, MDS finds the best-

fitting 2D coordinates that best match with the observed distances. In the resulting plot, nearby stimuli correspond to hard searches. The perceptual space for words and nonwords is shown in *Appendix 7—figure 1 A-B*. It can be seen that stimuli with common first letters are grouped together. MDS coordinates for nonwords was rotated without altering their overall configuration so as to best match the MDS coordinates for words.

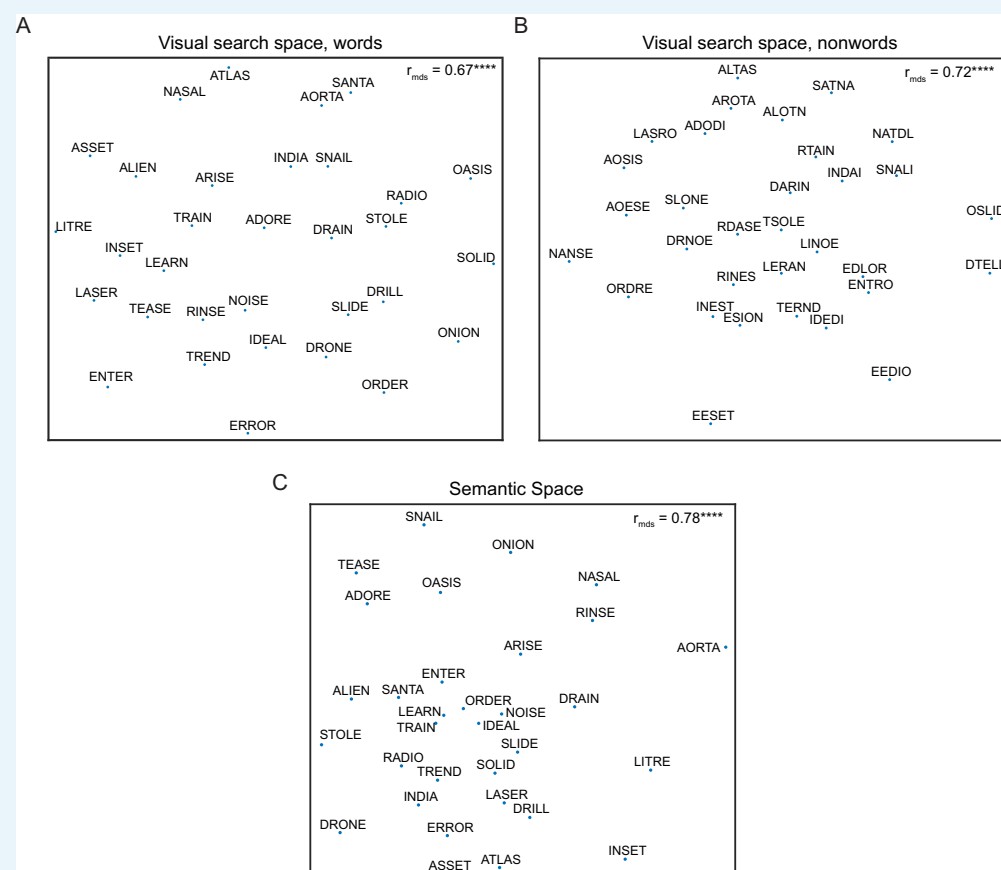

**Appendix 7—figure 1.** Multi-dimensional representation of words and nonwords. (**A**) Perceptual space for words. we used multidimensional scaling to find the 2D coordinates of all words that best match the observed distances. In the resulting plot, nearby words indicate hard searches. The correlation coefficient between dissimilarities in 2D plane and the observed data is shown. Asterisks indicate significant correlation (**** is p<0.00005). (**B**) Same as (**A**) but for nonwords. (**C**) Same as (**A**) but for semantic space of words.

The semantic dissimilarities were estimated using the GloVe features (*Pennington et al., 2014*), and visualized using MDS analysis (*Appendix 7—figure 1C*). In the resulting plot, semantically related words/frequently cooccurring words are closer to each other.

## Neural activity corresponding to words, nonwords, and letters

For each category of stimuli that is words, nonwords, and letters, we averaged the activity values across voxels and subjects within each ROI. The mean activity values are shown in *Appendix 7—figure 2A-E*. Since the activation levels can also be influenced by the reaction time (RT) in the lexical decision task, we regressed out the contributions of observed RT for each subject. Specifically, we estimated the contribution of RT by solving the linear equation $y = Xb$. Here $y$ is $64 \times 1$ vector of mean beta values, $X$ is $64 \times 2$ matrix containing the RTs in the first column and ones in the second column, and $b$ is a $2 \times 1$ vector of unknown model

coefficients. We estimated the vector **b** (estimated using the MATLAB function *regress*). Next, we subtracted the contribution of RT from the mean beta values and replotted the average activity values for words and nonwords (*Appendix 7—figure 2F*).

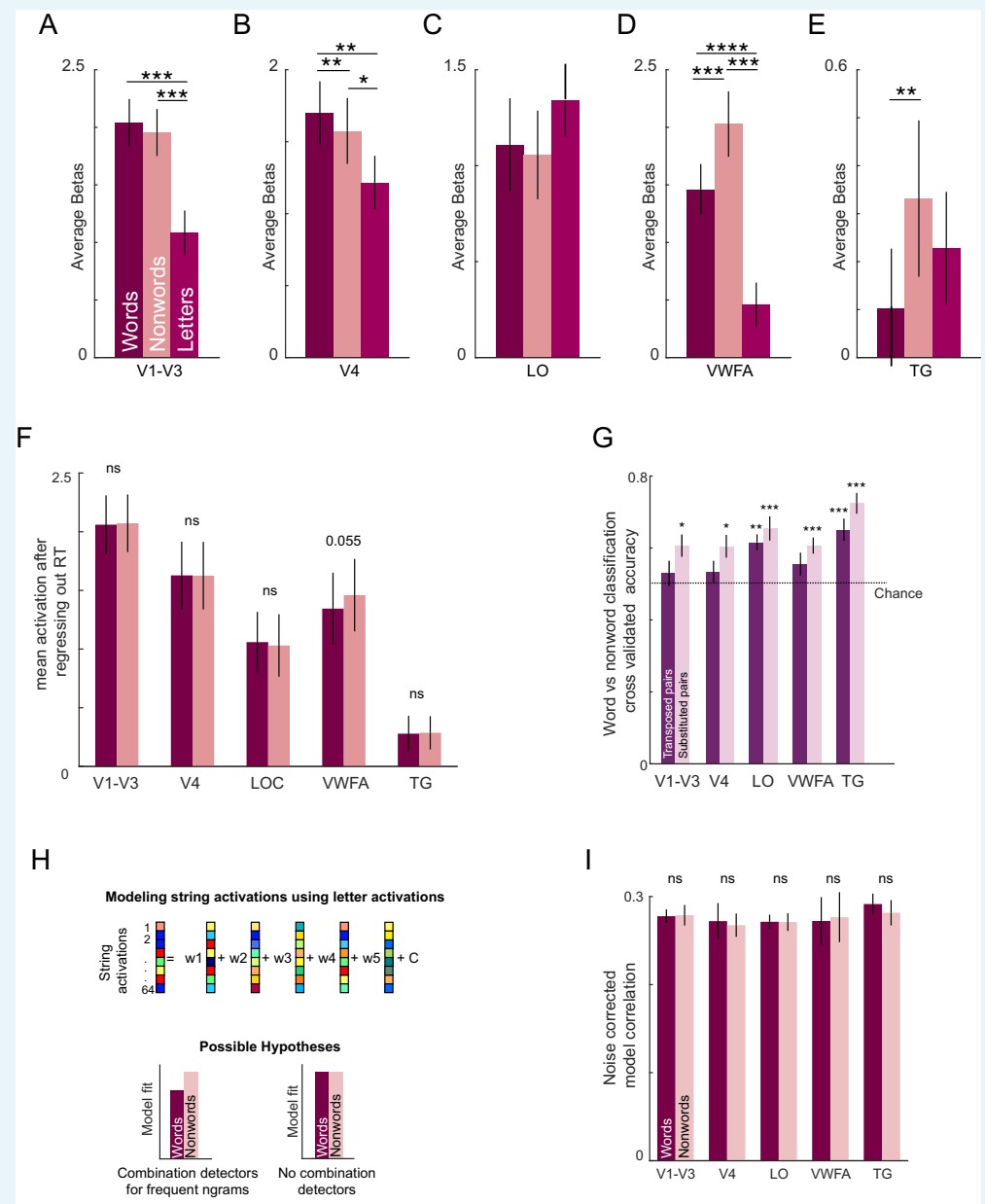

**Appendix 7—figure 2.** Neural activity. (**A**) Average activation levels for words, nonwords, and letters. Error bar indicate ±1 s.e.m. across subjects. Asterisks indicate statistical significance (* is p<0.05, ** is p<0.005, etc. in a sign-rank test comparing subject-wise average activations). (**B**)-(**E**). Same as in A but for V4, Lateral Occipital areas, Visual Word Form Area, and Temporal Gyri respectively. (**F**) Mean activation level after regressing out the reaction time in the lexical decision task. Error bars indicate ±1 s.e.m. across subjects. (**G**) Cross-validated classification accuracy for transposed word-nonword pairs (*dark*) and substituted word-nonword pairs (*light*). Error bars indicate s.e.m. across subjects. Asterisks indicate statistical significance (* is p<0.05, ** is p<0.005, etc. in a sign-rank test comparing subject-wise accuracy w.r.t. chance level). (**H**) Schematic of the voxel model. The response of each voxel across strings is modeled as a linear combination of the constituent letter responses.

Bottom: Hypothetical model fits based on the presence (right) or absence (left) of local combination detectors. Predicted responses for words will deviate from the observed responses under the influence of LCD. (**I**) Average model correlation (normalized using split-half correlation) for each ROI for words (dark) and nonwords (light). Error bar indicates s.e.m. across subject.

## Word vs nonword classification

For each ROI and subject, we built linear classifier to discriminate between words and nonwords (using the built-in MATLAB routine *fitcdiscr*). We built separate classifiers to distinguish the activity pattern of transposed and substituted nonwords from their corresponding word activity patterns. The resulting decoding accuracy is shown in *Appendix 7—figure 2G*. It can be seen that decoding accuracy for substituted nonwords is significantly better than for transposed nonwords (*Appendix 7—figure 2G*). Correspondingly, in behavior, subjects were faster at responding to substituted nonwords compared to transposed nonwords (response times, mean ± sd: 1.03 ± 0.08 s for 16 substituted nonwords, 1.20 ± 0.15 s for 16 transposed nonwords, p<0.005, rank-sum test comparing average response times).

## Can string responses be predicted from single letters?

We modeled the response of each voxel across the 64 strings (32 words, 32 nonwords) as a linear combination of the single letter activations (*Appendix 7—figure 2H*). We evaluated model fits by comparing model correlations separately for words and nonwords. If string responses were driven by specialized detectors for letter combinations (such as those present in words), then we reasoned that model correlations would be worse for words compared to nonwords. By contrast, if there are no specialized detectors of this kind, model fits would be equivalent for words and nonwords.

We calculated cross-validated model fits by training the model on half the trials and testing it on the other half of the trials. Since voxels could vary widely in their reliability of responses to the stimuli, we normalized the model fit of each voxel by its split-half reliability. The average noise-corrected model fit (averaged across voxels and subjects) is shown in *Appendix 7—figure 2I*. This revealed no systematic difference in model performance for words and nonwords in any of the ROIs (*Appendix 7—figure 2I*). We obtained qualitatively similar results using a searchlight, where there were no clear regions in which model fits differed for words and nonwords (*Appendix 7—figure 4D*).

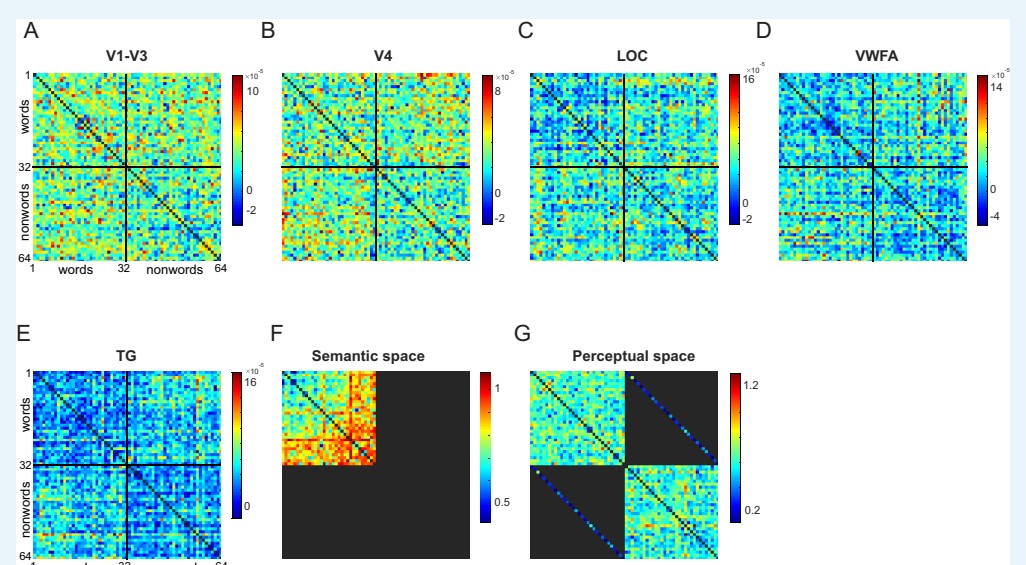

**Appendix 7—figure 3.** Representation dissimilarity matrix. (**A**) Average pair-wise dissimilarity values for all possible pairs of words (1-32) and nonwords (33-64). Colour bar represents dissimilarity values that were estimated using a cross-validated, normalized variation of Mahalanobis distance. (**B**) - (**E**) Same as in A but for V4, Lateral Occipital areas, Visual Word Form Area, and Temporal Gyri respectively. (**F**) Pair-wise dissimilarity values in the semantic space across all possible pairs of words. Color bar represents dissimilarity values that is 1 − r (correlation between feature vectors for a given word pair) (**G**) Average pair-wise perceptual dissimilarities (1/search reaction time) for all possible pairs of words, nonwords, and corresponding word-nonword pairs.

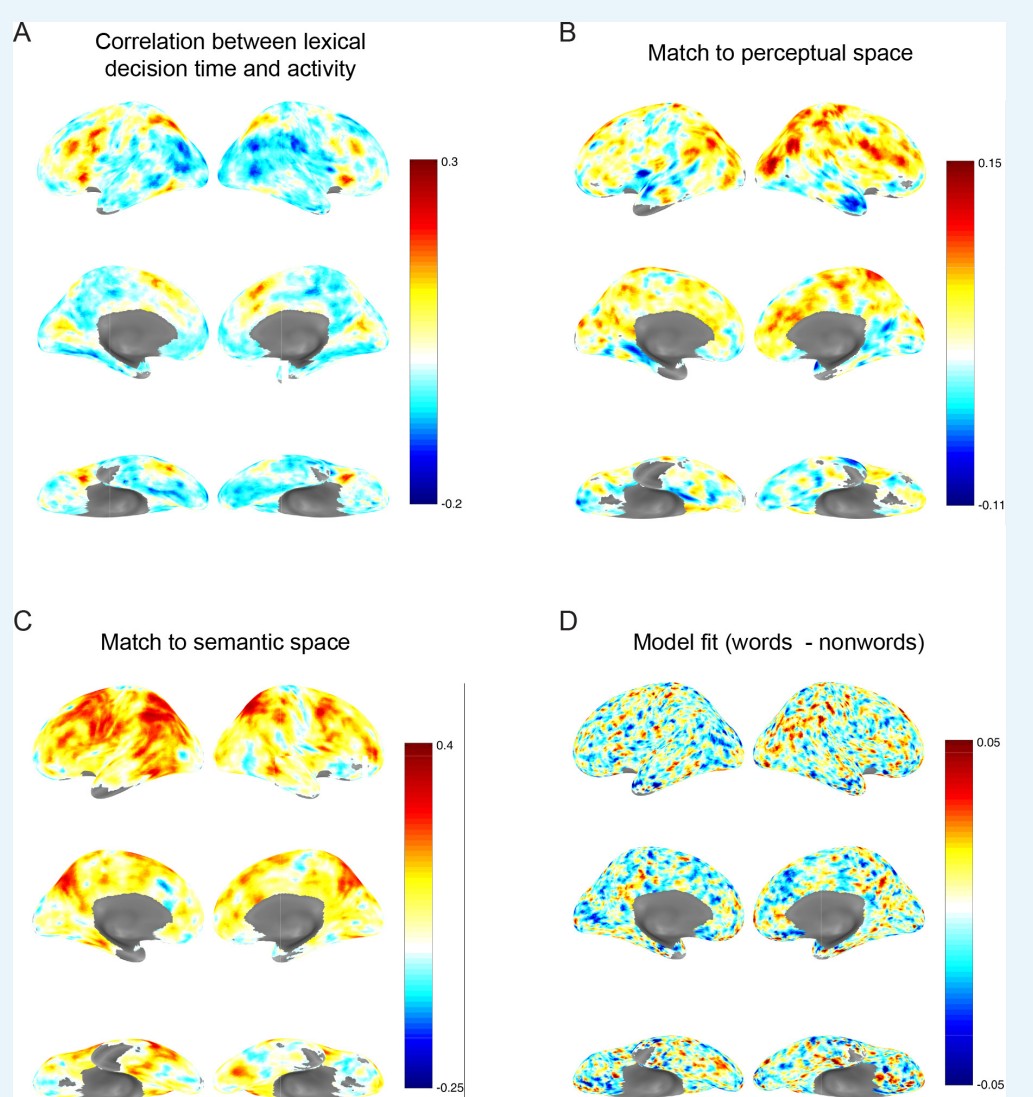

**Appendix 7—figure 4.** Searchlight analysis. (**A**) Searchlight map of correlation between neural activity and lexical decision time for each voxel. (**B**) Searchlight map of correlation between neural dissimilarity and search dissimilarities in behavior. (**C**) Searchlight map of correlation between neural dissimilarity and semantic dissimilarities. (**D**) Searchlight map depicting the difference in model fit for words versus nonwords for each voxel, averaged across subjects.

To further validate the letter model, we compared the single letter tuning along each MDS dimension with the observed single letter tuning in each ROI (*Appendix 7—figure 5A*). For each ROI, we grouped voxels with similar response profile and matched it to the MDS dimension (*Appendix 7—figure 5A*). We obtained similar single letter tuning and weight profiles for voxels across different ROIs. However, this analysis is inconclusive because there is no systematic way to compare a small set of neurons inferred from behavior with the much larger, possibly overcomplete set of voxel activations observed in brain imaging. Likewise, we grouped voxels with similar summation weights to compare the weight profiles in behavior and brain imaging. However this analysis is also inconclusive because different MDS-derived neurons might contribute differently towards behavior, so the summation weights cannot be directly averaged to make overall comparisons between ROI activations and behavior. Despite these caveats, there is a general match between tuning profiles and summation weights observed in behavior with those observed in different brain regions.

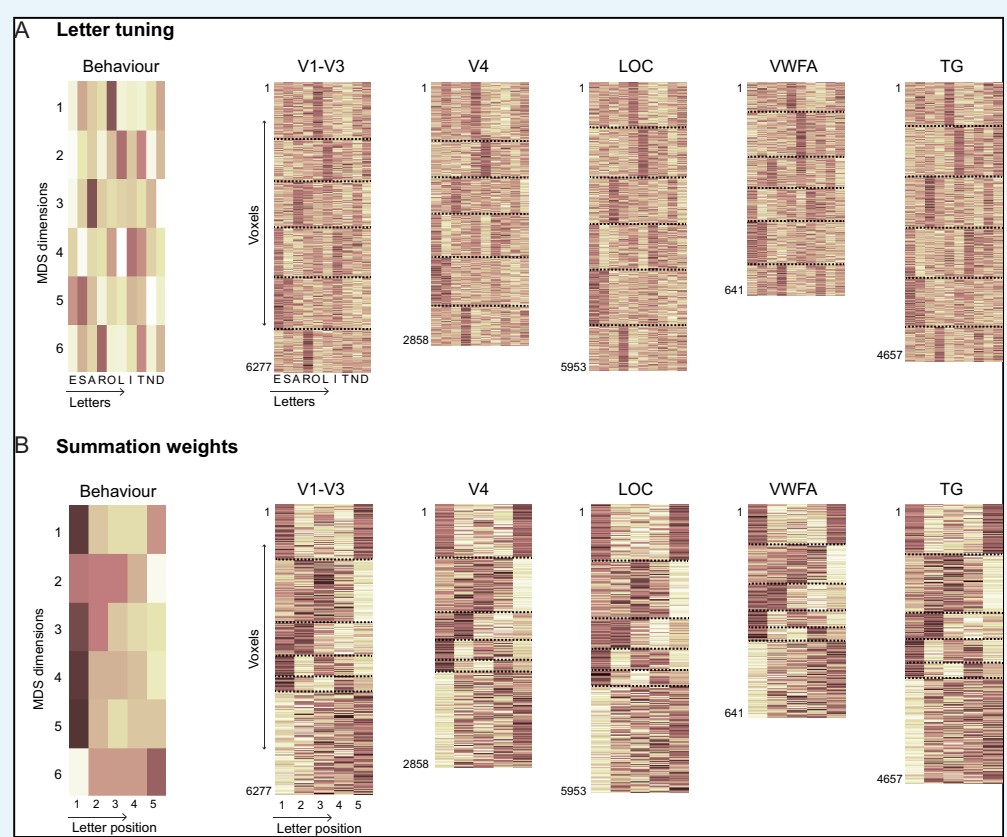

**Appendix 7—figure 5.** Comparison of letter tuning and summation weights. (**A**) (*Left*) Response of 6 MDS neurons for all the 10 letters. (*Right*) Single letters response across all the voxels (concatenated across subjects) within a given ROI. Each voxels is sorted into one of six groups depending on which MDS neuron it matches best. The height of each ROI plot is logarithmically scaled to match the number of voxels across all subjects. Black dashed lines are used to separate the clusters corresponding to each MDS neuron. (**B**) Same as (**A**) but showing the summation weights corresponding to each MDS neuron or ROI voxel.

## Neural dissimilarity values across words and nonwords

Within a given ROI, we calculated the pair-wise dissimilarity values using a noise-normalized variation of cross-validated Mahalanobis distance (*Nili et al., 2014*). The median dissimilarity matrix across all subjects is shown in *Appendix 7—figure 3A-E*. The semantic distance between every pair of words was computed as the cosine distance between the GloVe (*Pennington et al., 2014*) feature vectors activated by the two words (*Appendix 7—figure 3F*). The perceptual distance (1/search reaction time) averaged across all subjects for word-word pairs, nonword-nonword pairs, and word-nonword pair is shown in *Appendix 7—figure 3G*

## Searchlight analyses

To identify other brain regions that might show the effects observed in the individual ROIs, we performed a whole-brain searchlight analysis. Specifically, for each voxel in a given subjects' brain, we considered a local neighborhood of 27 voxels (3 × 3×3 voxels) and performed the following analyses of interest. We obtained similar results for larger searchlight volumes. The resulting maps were smoothed using a Gaussian filter with FWHM of 3 mm.

## Searchlight for regions that match lexical decision time

For each voxel, its activity across strings is correlated with mean lexical decision time. The resulting whole brain correlation map is averaged across subjects. Overall, activity in VWFA, Superior Parietal Lobe (SPL), Pre-Frontal and motor cortex was correlated with lexical decision time. This correlation map was visualized on the brain surface (*Appendix 7—figure 4A*).

## Searchlight for regions that match perceptual space

For the neighborhood of each voxel, we calculated the pairwise neural dissimilarity for all word-word, nonword-nonword, and word-nonword pairs for a given subject, and averaged this across subjects. We then calculated the correlation between this local neural dissimilarity and the corresponding string dissimilarities estimated using experiment 7. This correlation map was visualized on the brain surface (*Appendix 7—figure 4B*).

## Searchlight for regions that match semantic space

For the neighborhood of each voxel, we calculated the pairwise neural dissimilarity for all word-word pairs for a given subject and averaged this across subjects. We then calculated the correlation between this local neural dissimilarity and the corresponding semantic dissimilarities. This correlation map was visualized on the brain surface (*Appendix 7—figure 4C*).

## Searchlight for comparing linear model fits between words and nonwords

For each subject and voxel, we modeled the response to strings as a linear combination of its single letter responses. The model fits (correlation between observed and predicted string responses) was evaluated separately for words and nonwords. The difference in the mean model fits between words and nonword is visualized on the brain surface (*Appendix 7—figure 4D*).

## Match between letter model and fMRI data

The letter model described throughout the study is derived from dissimilarities measured in behavior in two steps. First, the dissimilarities between single letters were used to construct single neurons tuned to letter shape, whose activity predicts these dissimilarities. Second, the summation weights of each neuron were adjusted so that they match the dissimilarities between longer strings.

Given that we recorded responses to single letters as well as strings in fMRI, we wondered whether these can be matched in some manner to the letter tuning and summation weights derived from behavior in the letter model. Any direct comparison is fraught with the difficulty that many single letter tuning functions could produce the same behavior. For instance, simply rotating the MDS-derived tuning functions could yield another set of neurons that match the observed letter dissimilarities. This is further compounded by the fact that the MDS-derived neurons contribute unequally to behavior, and by the fact that this mapping could change completely with increasing numbers of neurons. Thus, it is unreasonable to expect voxel tuning for single letters or the summation weights to match exactly with the behaviorally derived tuning.

Nonetheless, we attempted to find a broad link between the single letter tuning and summation weights observed in behavior with those observed in each ROI. The results are summarized in *Appendix 7—figure 5A*. Since there are only 10 single letters, 6 MDS neurons were sufficient to explain >95% of the variance of the pair-wise single letter dissimilarities observed in Experiment 1. For each MDS neuron, we identified the voxels whose activity for single letters had the least residual error compared to other MDS neurons.

In this manner, we sorted the voxels into six groups corresponding to each MDS neuron. The resulting plots are shown in *Appendix 7—figure 5A*. It can be see that all ROIs show single letter tuning profiles similar to the behaviorally derived single letter tuning profiles. The corresponding summation weights for these voxels are shown in *Appendix 7—figure 5B*. Once again, it can be seen that many ROIs show similar summation weights as those observed in behavior.

