## [Decision Letter]

**Acceptance summary:**

The manuscript presents a detailed and in-depth study of word processing, providing evidence that a compositional letter code potentially makes a major contribution to word reading and can account for reading of jumbled words. One of the major strengths of this work is the combination of careful behavioral experiments, a simple neural model based on properties of neurons in monkey IT cortex, and fMRI data in human participants.

**Decision letter after peer review:**

Thank you for submitting your article "A compositional letter code in high-level visual cortex explains how we read jumbled words" for consideration by *eLife*. Your article has been reviewed by three peer reviewers, including Chris I Baker as the Reviewing Editor and Reviewer #1, and the evaluation has been overseen by Floris de Lange as the Senior Editor.

The reviewers have discussed the reviews with one another and the Reviewing Editor has drafted this decision to help you prepare a revised submission.

Summary:

The reviewers find the manuscript describes a very interesting and detailed study of the potential contribution of a compositional letter code to word reading using novel approaches across behavioral and neuroimaging data. A simple model of responses to individual letter shapes can explain performance in visual search tasks and capture some aspects of word and non-word processing. Further, neuroimaging data suggests that the compositional letter code corresponds to processing in a region in lateral occipital cortex, with the visual word form area (VWFA) corresponding to processing of properties related to word knowledge. One of the major strengths of this work is the combination of careful behavioral experiments, a simple neural model based on properties of neurons in monkey inferotemporal cortex, and fMRI data in human participants.

While the manuscript is well-written and comprehensive, the reviewers felt there are a number of revisions that would strengthen the work. These revisions would require significant reframing of the manuscript to more accurately reflect the data and the existing literature.

Essential revisions:

1) The conclusions tend to be overstated. The manuscript provides an interesting demonstration of what a visually based model can accomplish, but the authors go too far in their interpretation. The compositional letter code explains aspects of word and non-word processing, but not everything related to word reading. For example, in the lexical decision task, response times for words are explained by word frequency and there will be contributions from semantics, especially when reading strings of words/sentences. The authors should be clearer in describing what the data show and not over-reach into an account of word reading that discounts other factors without justification. This will also require a change in the title of the paper since most of the tasks the authors employ are not about word reading.

2) Related to point (1), the authors make a series of assertions that are not supported by their data and also present an oversimplified and somewhat biased review of the literature. The authors should clarify and justify these assertions. In particular:

a) "we hypothesized that word reading is enabled by a purely visual representation". There is a large and robust literature demonstrating how various linguistic properties have substantial effects on word recognition and the authors should acknowledge and cite this literature.

b) Visual search is very different than reading. It should be made clear that this task is not a test of the more general computations involved in reading.

c) "This difference in visual similarity explains why transposing the middle letters renders a word easier to read than transposing its edge letters". The authors should more clearly explain or qualify this assertion.

d) The two sentences in the Introduction seem to contradict each other. It is first suggested that word reading could be explained by a purely visual code, but then it is stated that reading could be a confound. It seems that what the authors want to say is that many different tasks involving words *can* (but are not necessarily always) be achieved by visual computations. This is a different claim than what is written throughout the paper.

e) The work as presented is not a test of the hypothesis that cortical regions involved in word recognition have tuning for letter combinations. Evidence for this hypothesis is based on a very different paradigm and the effect has been replicated by at least 3 different labs. The authors should clearly explain how the phenomenon they report aligns with those data.

f) Conceptually it is unclear how the bigram model lines up with the main assertion of the manuscript that performance can be understood in terms of the summation of performance for single letters. As soon as additional weights are added for bigrams isn't this akin to saying that there is a different representation of bigrams than of individual letters?

g) "According to an influential account of word reading…. ". This influential account of word reading is about the word recognition process not the detection of bigrams in a search array. It makes no comment or prediction about how the proposed neurons would be involved in the tasks developed by the authors. This is not to say that the model of bigram detectors in word-selective cortex is correct, just that the present work can be seen as orthogonal to this model and the authors need to clarify and justify how their data relates.

h) Subsection “Experiment 5: Lexical decision task” – why is the model retrained? If the model based on letters in general is used in this task shouldn't the same weights be used?

i) The summary sentence "In sum, we conclude that word response times are explained by word frequency and nonword response times are explained by the distance between the nonword and the nearest word calculated using the compositional neural code." accurately captures what the data show. It highlights how the authors' model makes interesting predictions in a variety of contexts. But it also contradicts the main assertions laid out in the Introduction as they note that many factors that are not purely visual explain a major portion of the variance.

j) Some of the findings on the fMRI task have been reported a number of times in the literature (e.g. the difference in VWFA response to words and pseudowords). The manuscript would be strengthened by relating it to these previous studies.

3) The authors find that LO is the region that is most sensitive to their visual similarity metric.

a) Given prior work showing the critical role of VWFA in reading, this result would seem to suggest that visual similarity is not at the core of reading per se and this issue should be discussed.

b) The paragraph beginning with "To further investigate the link between the compositional…" needs clarifying. Further, prior work has focused on the VWFA for the claim of bigram detectors and the authors needs to explain how the finding of a compositional code in LO relates to that work.

c) The authors also show that LO has a representation of semantic space. This suggests a contribution beyond visual properties. What does this mean for the role of these neurons? What are the implications of this overlap for theories of orthographic and semantic processing? This needs to be discussed.

4) Some results do not seem to support the ideas of the paper, but are framed as doing so. Specifically:

a) "To quantify this observation, we asked whether the model error for each bigram pair, calculated as the absolute difference between observed and predicted dissimilarity, covaried with the average bigram frequency of the two bigrams (for both frequent bigrams and words). […] We conclude that bigram search can be explained entirely using single neurons tuned to single letters."

The significant negative correlation does not seem to fit with the later statement that "model errors are not systematically different for frequent compared to infrequent bigram pairs". The authors should clarify this point in this paragraph, but also explain how this result fits with main message of the paper. There are a number of effects that are against the idea the main hypothesis "that word reading is enabled by a purely visual representation" and that the model "explained human performance in both visual search as well as word reading tasks." (Abstract). For instance, the effect of familiarity on asymmetric spatial summation (subsection “Experiment 3: Upright versus inverted bigrams”), and the frequency effect (Experiment 5).

b) If the claim is "there are no specialized detectors for letter combinations", it's not clear how Experiment 5 (where the same letters in different orders give different RTs) fits with this. This seems to strongly support a role for particular letter combinations.

5) In the analysis of mean VWFA activity, response times do not seem to be controlled. This is important because RTs might influence the magnitude of the β coefficients in a systematic way that does not directly correspond to the relevant processing. These should be included in the model.

6) In a number of places in the manuscript, the authors draw conclusions about how neurons are tuned, but without directly recording from neurons, it is not possible to draw this conclusion. For example, "Our main finding is that viewing a string activates a compositional letter code, consisting of neurons tuned to single letters whose response to longer strings is a linear sum of single letter responses".

7) It was mentioned that outliers in dissimilarity values across subjects were removed using built-in routines in MATLAB (isoutlier function, MATLAB R2018a). This is unusual and not typically seen in dissimilarity analyses. Are the findings the same without this procedure? How many outliers are removed? How is this threshold calculated/selected?

8) The manuscript mentions that cross-validation was performed but details on this are absent. In any case, pairwise dissimilarity calculations should be performed across runs (i.e., not correlating between items in the same run). This is particularly true because the trials are close together and influenced by the adjacent BOLD response.

9) It would be worth discussing how these findings relate to results (and associated theories) regarding whether individuals and specifically the VWFA process orthographic stimuli in a holistic versus part-based fashion (e.g., Carlos et al., 2019). More generally, behavioral responses to inverted words are relevant to this work, and worthy of more discussion (for how such findings are, or are not, consistent with this manuscript).

10) The authors report an exhaustive series of experiments and there is extensive supplementary material – while the authors should be commended for providing so much material, it does make the manuscript challenging to read at times, requiring the reader to jump around different parts of the text to fully understand the methods and results. The authors should include information such as the number of subjects in each experiment and key aspects of the stimuli (total number, how selected etc.), and protocol (e.g. timing of stimulus presentation, stimulus size etc.) in the main text. Further, there appears to be some inconsistency between the naming of sections. In the main text the authors refer the reader to, for example, "Section S7", but the supplementary material itself is labelled as "Section A7", and the tables and figures are numbered sequentially throughout the supplementary material such that the relevant figure in "Section A7" might be "Figure S16" – all very confusing. The authors should reconsider how they are structuring the supplementary material to make it more intuitively organized.

---

## [Author Response]

Essential revisions:1) The conclusions tend to be overstated. The manuscript provides an interesting demonstration of what a visually based model can accomplish, but the authors go too far in their interpretation. The compositional letter code explains aspects of word and non-word processing, but not everything related to word reading. For example, in the lexical decision task, response times for words are explained by word frequency and there will be contributions from semantics, especially when reading strings of words/sentences. The authors should be clearer in describing what the data show and not over-reach into an account of word reading that discounts other factors without justification. This will also require a change in the title of the paper since most of the tasks the authors employ are not about word reading.

Thank you for raising this concern. We do agree with you that effects related to word frequency are not compositional i.e. cannot be explained using single letters. We have now carefully reviewed the entire manuscript to avoid overstating our results. We have also changed the title to “A compositional neural code in high-level visual cortex enables jumbled word reading”, which we think is an accurate summary of our findings.

2) Related to point (1), the authors make a series of assertions that are not supported by their data and also present an oversimplified and somewhat biased review of the literature. The authors should clarify and justify these assertions. In particular:a) "we hypothesized that word reading is enabled by a purely visual representation". There is a large and robust literature demonstrating how various linguistic properties have substantial effects on word recognition and the authors should acknowledge and cite this literature.

We agree that linguistic properties have substantial effects on word recognition, and we have now modified the Introduction to make this point explicit. We have also reworded our hypothesis to make it clearer.

b) Visual search is very different than reading. It should be made clear that this task is not a test of the more general computations involved in reading.

We completely agree, and we have acknowledged this in the Introduction.

c) "This difference in visual similarity explains why transposing the middle letters renders a word easier to read than transposing its edge letters". The authors should more clearly explain or qualify this assertion.

We meant that FOGRET is easy to read as FORGET because it is visually similar to FORGET. By contrast OFRGET is harder to recognize since it is visually dissimilar to FORGET. We have now made this clear in the Introduction.

d) The two sentences in the Introduction seem to contradict each other. It is first suggested that word reading could be explained by a purely visual code, but then it is stated that reading could be a confound. It seems that what the authors want to say is that many different tasks involving words can (but are not necessarily always) be achieved by visual computations. This is a different claim than what is written throughout the paper.

We meant that visual search for strings might have invoked some reading process, in which case explaining reading phenomena using visual search would be circular. To overcome this confound we have to demonstrate that visual search for strings can be explained using a purely visual model that does not include any lexical or linguistic factors, which is what we have done in our study. We have now reworked this paragraph to make it clearer (Introduction).

e) The work as presented is not a test of the hypothesis that cortical regions involved in word recognition have tuning for letter combinations. Evidence for this hypothesis is based on a very different paradigm and the effect has been replicated by at least 3 different labs. The authors should clearly explain how the phenomenon they report aligns with those data.

We now discuss these studies in relation to ours in the Discussion.

f) Conceptually it is unclear how the bigram model lines up with the main assertion of the manuscript that performance can be understood in terms of the summation of performance for single letters. As soon as additional weights are added for bigrams isn't this akin to saying that there is a different representation of bigrams than of individual letters?

Our letter-based model in figure only assumes neurons tuned for single letters and summation weights that depend on letter position, and therefore contains no information specific to any particular bigram. We have clarified this in the Results.

g) "According to an influential account of word reading…. ". This influential account of word reading is about the word recognition process not the detection of bigrams in a search array. It makes no comment or prediction about how the proposed neurons would be involved in the tasks developed by the authors. This is not to say that the model of bigram detectors in word-selective cortex is correct, just that the present work can be seen as orthogonal to this model and the authors need to clarify and justify how their data relates.

Actually there have been several proposals in the literature about open bigrams (Grainger and Whitney, 2004 ) and local combination detectors (Dehaene et al., 2005) according to which reading is enabled by detectors of higher order combinations of letters. Our model does stand in contrast to such proposals since it does not assume any tuning for letter combinations, yet it explains many aspects of orthographic processing. We have now reworked the text to make this clear (Introduction).

h) Subsection “Experiment 5: Lexical decision task” – why is the model retrained? If the model based on letters in general is used in this task shouldn't the same weights be used?

We retrained the model based on our observation that the spatial summation weights varied across the visual search experiments, presumably reflecting differing attentional resources across letter position. We have now updated the text to make this clear.

To assess whether the model trained on visual search data would also be able to predict nonword response times, we took the model trained on the visual search data in Experiment 4, and calculated the word-nonword distances using this model. This too yielded a significant positive correlation (r = 0.39, p < 0.00005) that was better than the OLD and lexical models. We have now included this in the Results.

i) The summary sentence "In sum, we conclude that word response times are explained by word frequency and nonword response times are explained by the distance between the nonword and the nearest word calculated using the compositional neural code." accurately captures what the data show. It highlights how the authors' model makes interesting predictions in a variety of contexts. But it also contradicts the main assertions laid out in the Introduction as they note that many factors that are not purely visual explain a major portion of the variance.

Thank you for this observation. We have revised the text throughout to more closely reflect these points.

j) Some of the findings on the fMRI task have been reported a number of times in the literature (e.g. the difference in VWFA response to words and pseudowords). The manuscript would be strengthened by relating it to these previous studies.

We have now reworked the Discussion to include these points.

3) The authors find that LO is the region that is most sensitive to their visual similarity metric.a) Given prior work showing the critical role of VWFA in reading, this result would seem to suggest that visual similarity is not at the core of reading per se and this issue should be discussed.

Thank you for raising this point. We have now included these points in the Discussion.

b) The paragraph beginning with "To further investigate the link between the compositional…" needs clarifying. Further, prior work has focused on the VWFA for the claim of bigram detectors and the authors needs to explain how the finding of a compositional code in LO relates to that work.

We now have reworked this paragraph to make it clearer (subsection “Neural basis of perceptual space”). We now discuss our findings in relation to previous studies on VWFA (subsection “Neural basis of word recognition”).

c) The authors also show that LO has a representation of semantic space. This suggests a contribution beyond visual properties. What does this mean for the role of these neurons? What are the implications of this overlap for theories of orthographic and semantic processing? This needs to be discussed.

We now acknowledge these points in the Discussion.

4) Some results do not seem to support the ideas of the paper, but are framed as doing so. Specifically:a) "To quantify this observation, we asked whether the model error for each bigram pair, calculated as the absolute difference between observed and predicted dissimilarity, covaried with the average bigram frequency of the two bigrams (for both frequent bigrams and words). […] We conclude that bigram search can be explained entirely using single neurons tuned to single letters."The significant negative correlation does not seem to fit with the later statement that "model errors are not systematically different for frequent compared to infrequent bigram pairs". The authors should clarify this point in this paragraph, but also explain how this result fits with main message of the paper.

Thank you for drawing our attention to this point. This was a weak negative correlation and we were concerned that it may not be real. Upon carefully analysing the data and model fits, we noticed that the weak correlation was not robustly significant across slightly different random starting points in the model fits. However the comparison between the top 20 vs. bottom 20 frequent bigrams was more robust and therefore we have included only this analysis and removed the correlation part.

There are a number of effects that are against the idea the main hypothesis "that word reading is enabled by a purely visual representation" and that the model "explained human performance in both visual search as well as word reading tasks." (Abstract). For instance, the effect of familiarity on asymmetric spatial summation (subsection “Experiment 3: Upright versus inverted bigrams”), and the frequency effect (Experiment 5).

We agree with you in general. We have now carefully reviewed the text throughout to more accurately reflect our findings. However, the effect of familiarity could be purely visual in nature and not necessarily due to linguistic factors, but our study does not distinguish between these possibilities. We have acknowledged this point in the Results.

b) If the claim is "there are no specialized detectors for letter combinations", it's not clear how Experiment 5 (where the same letters in different orders give different RTs) fits with this. This seems to strongly support a role for particular letter combinations.

No, we disagree. It would imply a role for particular letter combinations only if letters were equally important at every letter position in the word. Our model, which is based on tuning for single letters and with weights for each letter position, is able to predict the variation in nonword RT across different scrambled versions of a word (Figure 4). The fact that the spatial summation weights in the model are different at each letter position challenges this implicit assumption that all letters contribute equally.

5) In the analysis of mean VWFA activity, response times do not seem to be controlled. This is important because RTs might influence the magnitude of the β coefficients in a systematic way that does not directly correspond to the relevant processing. These should be included in the model.

We have now included an analysis of mean activations of each ROI after regressing out the word and nonword RTs from the activations (Appendix 7—figure 2). However, any generic effect of longer response times on mean activations would have produced artefactual correlations in all regions, but we observed a correlation only in the VWFA (Figure 5F).

6) In a number of places in the manuscript, the authors draw conclusions about how neurons are tuned, but without directly recording from neurons, it is not possible to draw this conclusion. For example, "Our main finding is that viewing a string activates a compositional letter code, consisting of neurons tuned to single letters whose response to longer strings is a linear sum of single letter responses".

That’s true. We have now reworked the text now to make this clearer.

7) It was mentioned that outliers in dissimilarity values across subjects were removed using built-in routines in MATLAB (isoutlier function, MATLAB R2018a). This is unusual and not typically seen in dissimilarity analyses. Are the findings the same without this procedure? How many outliers are removed? How is this threshold calculated/selected?

Yes, we obtained qualitatively similar without this step. Excluding outliers increased the split-half consistency of the data. The ‘isoutlier’ function by default removes any value greater than three scaled absolute deviations away from the median and was applied to each dissimilarity pair separately. This step removed 12.3% of the dissimilarity data. The default threshold was used here. We have now updated the Materials and methods to include these points.

8) The manuscript mentions that cross-validation was performed but details on this are absent. In any case, pairwise dissimilarity calculations should be performed across runs (i.e., not correlating between items in the same run). This is particularly true because the trials are close together and influenced by the adjacent BOLD response.

We have now included these details in the manuscript.

9) It would be worth discussing how these findings relate to results (and associated theories) regarding whether individuals and specifically the VWFA process orthographic stimuli in a holistic versus part-based fashion (e.g., Carlos et al., 2019). More generally, behavioral responses to inverted words are relevant to this work, and worthy of more discussion (for how such findings are, or are not, consistent with this manuscript).

While behavioural and neural responses to inverted words during lexical decisions have indeed been studied, they relate to recognizing familiar objects in novel orientations. Our finding relates to how part summation happens in upright versus inverted orientations, and therefore seemed somewhat unrelated. However, the finding that VWFA responds more strongly to inverted compared to upright words during lexical decisions is consistent with our finding that difficult nonwords activate VWFA strongly. We have now acknowledged this in the Discussion.

10) The authors report an exhaustive series of experiments and there is extensive supplementary material – while the authors should be commended for providing so much material, it does make the manuscript challenging to read at times, requiring the reader to jump around different parts of the text to fully understand the methods and results. The authors should include information such as the number of subjects in each experiment and key aspects of the stimuli (total number, how selected etc.), and protocol (e.g. timing of stimulus presentation, stimulus size etc.) in the main text. Further, there appears to be some inconsistency between the naming of sections. In the main text the authors refer the reader to, for example, "Section S7", but the supplementary material itself is labelled as "Section A7", and the tables and figures are numbered sequentially throughout the supplementary material such that the relevant figure in "Section A7" might be "Figure S16" – all very confusing. The authors should reconsider how they are structuring the supplementary material to make it more intuitively organized.

Thank you for these comments. We have included more details of each experiment in the main text. We have also carefully reviewed the text and supplementary material to make them more consistent. Finally, figures in the appendix are now numbered according to each section (e.g. Section A7 contains Appendix 7—figures 1, 2 etc).